# Past Evolution of Western Europe Large-scale Circulation and Link to Precipitation Trend in the Northern French Alps

Antoine Blanc[1], Juliette Blanchet[1], and Jean-Dominique Creutin[1]

[1]Univ. Grenoble Alpes, CNRS, IRD, Grenoble INP, IGE, 38000 Grenoble, France

**Correspondence:** Antoine Blanc (antoine.blanc2@univ-grenoble-alpes.fr)

**Abstract.** Detecting trends in regional large-scale circulation (LSC) is an important challenge as LSC is a key driver of local weather conditions. In this work, we investigate the past evolution of Western Europe LSC based on the 500 hPa geopotential height fields from 20CRv2c (1851-2010), ERA20C (1900-2010) and ERA5 (1950-2010) reanalyses. We focus on the evolution of large-scale circulation characteristics using three atmospheric descriptors that are based on analogy, by comparing daily geopotential height fields to each other. They characterize the stationarity of geopotential shape and how well a geopotential shape is reproduced in the climatology. A non-analogy descriptor is also employed to account for the intensity of the centers of action. We then combine the four atmospheric descriptors with an existing weather pattern classification over the period 1950-2019 to study the recent changes in the two main atmospheric influences driving precipitation in the Northern French Alps. They correspond to the Atlantic circulation pattern dominated by a zonal flow, and the Mediterranean circulation pattern dominated by low pressure anomalies over the near Atlantic, close to Portugal. Even though LSC characteristics and trends are consistent among the three reanalyses after 1950, we find major differences between 20CRv2c and ERA20C from 1900 to 1950 in accordance with previous studies. Notably, ERA20C produces flatter geopotential shapes in the beginning of the 20th century and shows a reinforcement of the meridional pressure gradient that is not observed in 20CRv2c. Over the period 1950-2019, we show that winter Atlantic circulations (zonal flows) tend to be shifted northward and they become more similar to known Atlantic circulations. Mediterranean circulations tend to become more stationary, more similar to known Mediterranean circulations and associated with stronger centers of action in autumn, while an opposite behaviour is observed in winter. Finally, we discuss the role of these LSC changes for seasonal and extreme precipitation in the Northern French Alps. We show that these changes in LSC characteristics are linked to (a) the decreasing contribution of Mediterranean circulations to winter precipitation, (b) more circulations that are likely to generate extreme precipitation in autumn.

## 1 Introduction

By defining the direction and intensity of airflow, modifying stability and moisture availability, large-scale circulation (LSC) is a key driver of local weather conditions. LSC variability over the Euro-Atlantic sector influences precipitation anomalies over Europe and the Mediterranean region. The North Atlantic Oscillation (NAO) is the first mode of LSC variability over the North Atlantic (Barnston and Livezey, 1987). In winter, a positive phase of NAO drives positive precipitation anomalies over Northern Europe and negative precipitation anomalies over Southern Europe by intensifying westerlies (Hurrell, 1995).

Nevertheless, other modes of LSC variability – such as the Euro-Atlantic blocking (EAB) or the East Atlantic/Western Russia pattern (EA/WR) – better explain precipitation variability in Central Europe, especially in the Alpine region which acts as a climatological barrier at the crossroad of different atmospheric influences (Auer et al., 2007; Bartolini et al., 2009; Beniston, 2005; Quadrelli et al., 2001; Scherrer et al., 2016). The Northern flanks of the Alpine range experience wet conditions under the Atlantic ridge pattern while the Southern flanks experience wet conditions when low pressure anomalies stand over the near Atlantic (Kotsias et al., 2019; Plaut and Simonnet, 2001).

Specific LSC patterns also drive extreme weather events over Europe and the Mediterranean region, including extreme precipitation (Pasquier et al., 2019), floods (Stucki et al., 2012) or extreme snowfall (Scherrer and Appenzeller, 2006). The probability of extreme precipitation events over Europe is decreased under a blocking high pressure system while it is increased southeast and southwest of the block (Lenggenhager and Martius, 2019). Extreme precipitation in the Northwestern, Northern and Central Alps are associated with low amplitude trough over the UK, zonally oriented flows and East Atlantic ridge driving southwesterlies-to-northwesterlies towards the region (Blanchet et al., 2021b; Giannakaki and Martius, 2016; Horton et al., 2012; Plaut et al., 2001). Extreme precipitation in the southern slopes of the Alps mainly occur in autumn and they are associated with low pressure systems developing from the near Atlantic to the Iberic Peninsula, driving southwesterlies and strong southerly flows (Blanchet et al., 2021b; Horton et al., 2012; Mastrantonas et al., 2021; Plaut et al., 2001).

Knowing the link between LSC and precipitation variability and extremes, changes in LSC may have significant impacts on local climate. Over the last $10,000$ years, increasing flood frequency in the European Alps cold periods may be linked to a southward shift of the Hadley cell and a weakening of the Açores high allowing a southward shift of the Westerlies and meandering circulations (Glur et al., 2013; Wirth et al., 2013). More recently, the flood-rich period in Central Europe in the 19th century appears to be associated with a more zonal and southward-shifted circulation (Brönnimann et al., 2019). Using LSC classification, Iannuccilli et al. (2021) show that part of the increase in extreme precipitation over Central Italy in winter and spring can be explained by changes in occurrence of the circulation types. In Southern France, the decreasing autumn and winter precipitation from $1951$ to $2000$ appears to be explained by a decrease in the occurrence of weather types driving precipitation over the region (Boé and Terray, 2008).

The aforementioned papers mainly deal with classes of LSC. However recent studies have pointed out the link between intrinsic characteristics of Western Europe LSC and precipitation variability and extremes in the Northern French Alps (Blanc et al., 2022, 2021; Blanchet et al., 2018; Blanchet and Creutin, 2020). Using atmospheric descriptors characterizing the daily $500$ hPa geopotential height field, the authors showed that LSC driving extreme precipitation in the Northern French Alps feature among the strongest centers of action with flow directions that are quite stationary and closely reproduced in the climatology – characteristics that are rare in the climatology (Blanc et al., 2022; Blanchet et al., 2018; Blanchet and Creutin, 2020). Furthermore, the reproducibility of flow direction and the intensity of the centers of action have been shown to perform as well as a weather pattern classification to explain seasonal precipitation in the Northern French Alps (Blanc et al., 2021). Seasons featuring LSC that are closely reproduced in the climatology or that feature strong centers of action tend to feature large precipitation accumulation, from autumn to spring.

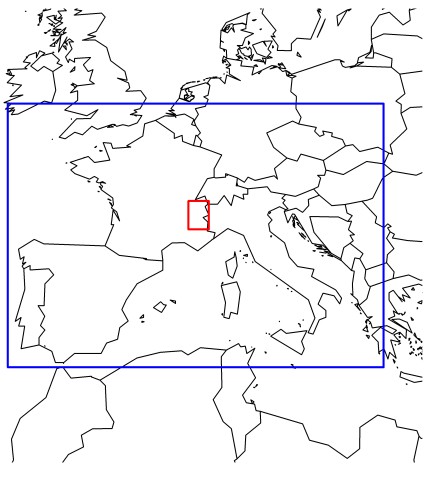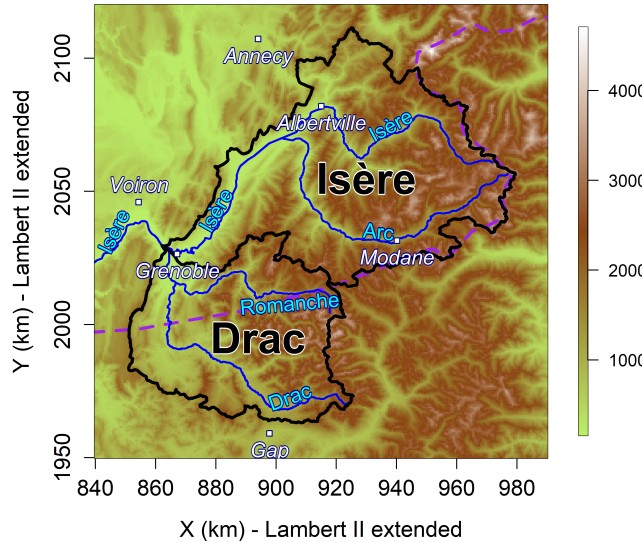

**Figure 1.** Left: Map of Western Europe with the Northern French Alps (red) and the domain considered for the 500 hPa geopotential height (blue). Right: Altitudinal map (in meters) of the Isère and the Drac River catchments at Grenoble, with the main cities and rivers. The purple dotted line represents the climatological border of Auer et al. (2007). Coordinates projection is according to the Lambert II extended system.

Given the link between LSC intrinsic characteristics and precipitation, this paper studies the past evolution of the atmospheric descriptors introduced in Blanc et al. (2022, 2021); Blanchet et al. (2018); Blanchet and Creutin (2020) and its link to precipitation trend in the Northern French Alps. The atmospheric descriptors allow assessing trend in daily LSC rather than trend in weather pattern occurrence at the seasonal scale. The four atmospheric descriptors are first employed to study the long-term evolution of Western Europe LSC from 1851 to 2010 using different reanalyses products. They are then combined with an existing weather pattern classification over the period 1950-2019 to address recent changes in the main atmospheric influences driving precipitation in the Northern French Alps. Finally, trends in seasonal precipitation and extreme precipitation in the Northern French Alps are considered over the period 1950-2017 and the role of LSC changes is discussed.

## 2 Data

### 2.1 Large-scale circulation

We use daily 500 hPa geopotential height fields over a $32° \times 16°$ region to represent Western Europe LSC (blue rectangle in Fig. 1). The 500 hPa geopotential height ranges from $4,800$ m to $6,100$ m, giving information about the pressure distribution in the middle of the troposphere. We extracted the 500 hPa geopotential height from three different reanalyses covering different periods (Table 1).

| Name | Institution | Period | Horizontal resolution | Model | Assimilated data |
|---|---|---|---|---|---|
| 20CRv2c | NOAA-CIRES | 1851-2014 | $2° \times 2°$ | GFS-2008ex | surface-input (surface pressure) |
| ERA20C | ECMWF | 1900-2010 | $1.125° \times 1.125°$ | IFS-Cy38r1 | surface-input (surface pressure, marine wind) |
| ERA5 | ECMWF | 1950-today | $0.25° \times 0.25°$ | IFS-Cy41r2 | full-input |

**Table 1.** Main properties of the reanalyses used.

We use the 20CRv2c reanalysis from NOAA-CIRES (Compo et al., 2011). 20CRv2c provides information about the state
of the atmosphere since 1851 with an horizontal resolution of $2°$. It only assimilates surface pressure observations using an
Ensemble Kalman Filter, and it is composed of 56 individual members that are equiprobable as well as a mean member. Sea
surface temperature and sea-ice distributions are used as boundary conditions. The Ensemble Kalman Filter allows considering
time varying observational uncertainty. The standard deviation in sea level pressure over the North Atlantic between 20CRv2c
members has been shown to be larger in the beginning of the reanalysis than in the recent period, affecting the mean member
homogeneity over time (Rodrigues et al., 2018). In this article, we use two individual members – arbitrary member 1 and
member 2 – as well as the mean member to determine whether significant differences are observed between the individual and
the mean members.

The twentieth century reanalysis ERA20C from ECMWF is also used (Poli et al., 2016). ERA20C provides a higher spatial
resolution than 20CRv2c with a $1.125°$ grid, but it ranges over a shorter period (1900-2010). In addition to surface pressure,
ERA20C also assimilates marine wind observations using a 4D-Var assimilation technique. ERA20C is single-member. It is
forced by sea surface temperature, sea-ice cover, atmospheric composition and solar forcing.

Finally, we use the ERA5 reanalysis which is the most recent reanalysis product of ECMWF (Hersbach et al., 2020). ERA5
ranges over a more recent period (1950-today) but it provides atmospheric variables with a high spatial resolution of $0.25°$.
ERA5 assimilates surface observations, upper-air observations and satellite observations (referred to as "full-input") using
a 4D-Var scheme. ERA5 relies on the radiative forcing of CMIP5 including total solar irradiance, ozone, greenhouse gases
and some aerosols including stratospheric sulfate aerosols. It takes sea-surface temperature and sea-ice cover as boundary
conditions. A 10-member ensemble with reduced resolution is available, but we use the high resolution realisation of ERA5
which is referred to as the main reanalysis.

## 2.2 Precipitation

We use the SPAZM precipitation data set in the Northern French Alps over the period 1950–2017 (Gottardi et al., 2012).
SPAZM is a $1 \times 1$ km$^2$ gridded interpolation of daily rainfall accumulations measured by more than $1,800$ daily rain gauges
over the French Alps, the Pyrenees, the Massif Central, Switzerland, Spain and Italy. SPAZM decomposes the rainfall field
into a guess field incorporating orography and residuals to this field. A distinctive feature of SPAZM is that the guess field is

conditional on the weather pattern of the target day, using the weather pattern classification of Garavaglia et al. (2010). This allows adjusting the altitudinal gradients according to the interaction between air flow and mountainous areas.

Following the works of Blanc et al. (2022), we focus on two catchments of the Northern French Alps, namely the Isère and the Drac River catchments at Grenoble. The Isère River catchment at Grenoble sizes $5,800$ km$^2$, with altitude ranging from $200$ m in Grenoble to more than $3,800$ m in the Vanoise massif (Fig. 1). The Drac River catchment at Grenoble sizes $3,600$ km$^2$, with altitude ranging from $200$ m in Grenoble to $4,100$ m in the Ecrins massif. The Ecrins massif is known as a climatological barrier between the Northern and the Southern French Alps, as shown by the climatological subregions of Auer et al. (2007) (see Fig. 1). As a consequence, the Isère River catchment is mainly affected by moisture coming from the Atlantic, while the Drac River catchment is mainly affected by moisture coming from both the Atlantic and the Mediterranean. The interannual catchment precipitation in both the Isère and the Drac River catchment reaches $1,300$ mm/year.

## 3    Method

Changes in LSC are studied using atmospheric descriptors characterizing daily $500$ hPa geopotential height fields. An existing weather pattern classification is also employed to consider changes in LSC characteristics that are specific to the main atmospheric influences driving precipitation in the Northern French Alps.

### 3.1    Atmospheric descriptors

Changes in LSC characteristics are investigated using four atmospheric descriptors introduced in previous works (Blanc et al., 2022; Blanchet et al., 2018; Blanchet and Creutin, 2020). These descriptors are based on daily $500$ hPa geopotential height fields over Western Europe (rectangle in Fig. 1). Three descriptors are based on analogy, by comparing daily geopotential height fields to each other. The analogy is based on the Teweles-Wobus score (Teweles and Wobus, 1954), which measures the similarity in shape between geopotential height fields. The geopotential shape defines the flow direction which is relevant for regional precipitation. The Teweles-Wobus score (TWS) is therefore widely used in the analog method to reconstruct precipitation based on geopotential height fields (Daoud et al., 2016; Marty et al., 2012; Wetterhall et al., 2005). The direction of the flow is even more relevant for precipitation in mountains, which act as climatological barriers with different flow directions impacting neighboring regions (Auer et al., 2007; Blanchet et al., 2021b; Horton et al., 2012). The TWS between days $t_k$ and $t_{k'}$ is given by:

$$TWS_{k,k'} = \frac{\sum_{(j,j') \in Adj} |(z_{jk} - z_{j'k}) - (z_{jk'} - z_{j'k'})|}{2 \sum_{(j,j') \in Adj} \max(|z_{jk} - z_{j'k}|, |z_{jk'} - z_{j'k'}|)}, \tag{1}$$

where $Adj$ ranges the set of pairs of adjacent grid points in the eastern and northern direction in the domain. $TWS_{k,k'}$ represents a normalized sum of differences in meridional and zonal gradients at all gridpoints over Western Europe between the $500$ hPa geopotential height field of day $k$ and day $k'$. A $TWS_{k,k'}$ of $0$ means that day $k$ and $k'$ feature strictly identical flow directions. A $TWS_{k,k'}$ of $1$ means that day $k$ and $k'$ feature strictly opposite flow directions. In practice, the TWS obtained in this study range between $0.04$ and $0.88$. Furthermore, it is important to have in mind that the TWS can produce large scores –

reflecting different flow directions – for quite flat geopotential shapes, representing weak differences in absolute flow, due to the normalization in Eq. (1).

Fig. 2a provides a schematic illustration of the three descriptors based on analogy in flow direction. The first descriptor is the celerity that is understood as the speed of deformation of the geopotential. It measures the stationarity in flow direction between two consecutive days. It is defined for day $t_k$ as the TWS between day $t_k$ and day $t_{k-1}$ (dotted arrow in Fig. 2a):

$$cel_k = TWS_{k-1,k}. \tag{2}$$

The lower the celerity, the more stationary the flow direction between two consecutive days.

The two other descriptors based on analogy are the singularity and relative singularity. They measure the way a flow direction is reproduced in the climatology. The singularity and relative singularity rely on the comparison of a given flow direction with its $Q$ closest flow directions in the climatology, referred as its analogs. The singularity of day $t_k$ is defined as the mean TWS between day $t_k$ and its $Q$ closest analog days (mean of solid arrows in Fig. 2a):

$$sing_k = \frac{1}{Q} \sum_{q \in A_k} TWS_{k,q}, \tag{3}$$

where $\mathcal{A}_k$ range the $Q$ closest analogs of day $t_k$. A flow direction featuring a low singularity means that close flow directions are found in the climatology. The singularity cannot be directly related to the frequency of occurrence of a given flow direction since a geopotential shape is never perfectly reproduced ($TWS_{k,k'} > 0$). Very low singularities even appear to be rare in the climatology, which means that the atmosphere spends much time exploring quite unseen patterns than very closely coming back to an already seen pattern (Blanc et al., 2022; Blanchet and Creutin, 2020).

The relative singularity of day $t_k$ is defined as the singularity normalized by the Teweles-Wobus score with the $Qth$ closest analog day (mean of solid arrows normalized by the dashed arrow in Fig. 2b):

$$rsing_k = \frac{sing_k}{TWS_{k,(Q)}}. \tag{4}$$

The relative singularity measures the similarity of a given flow direction to its very close analogs in comparison to the farther analog. It measures in a way the degree of clustering of the closest flow directions. The relative singularity is closely related to the local dimension of Faranda et al. (2017a) although they employ an Euclidean distance instead of TWS. A flow direction featuring a low singularity and relative singularity is said to be almost similarly reproduced in the climatology, as close flow directions are found in the climatology (low singularity) but the closest flow directions tend to be even more similar than usual (low relative singularity).

Blanchet and Creutin (2020) showed that the singularity and relative singularity are not very sensitive to the exact number of days selected as analog in Eq. (3) and Eq. (4), and that the selection of the closest 0.5 % days was a reasonable choice to link LSC characteristics with 3-day precipitation in the Northern French Alps. The period 1950-2010 is considered for the search of analog, as it is the common period of 20CRv2c, ERA20C and ERA5. Therefore, we use in the rest of this study $Q = 111$ days.

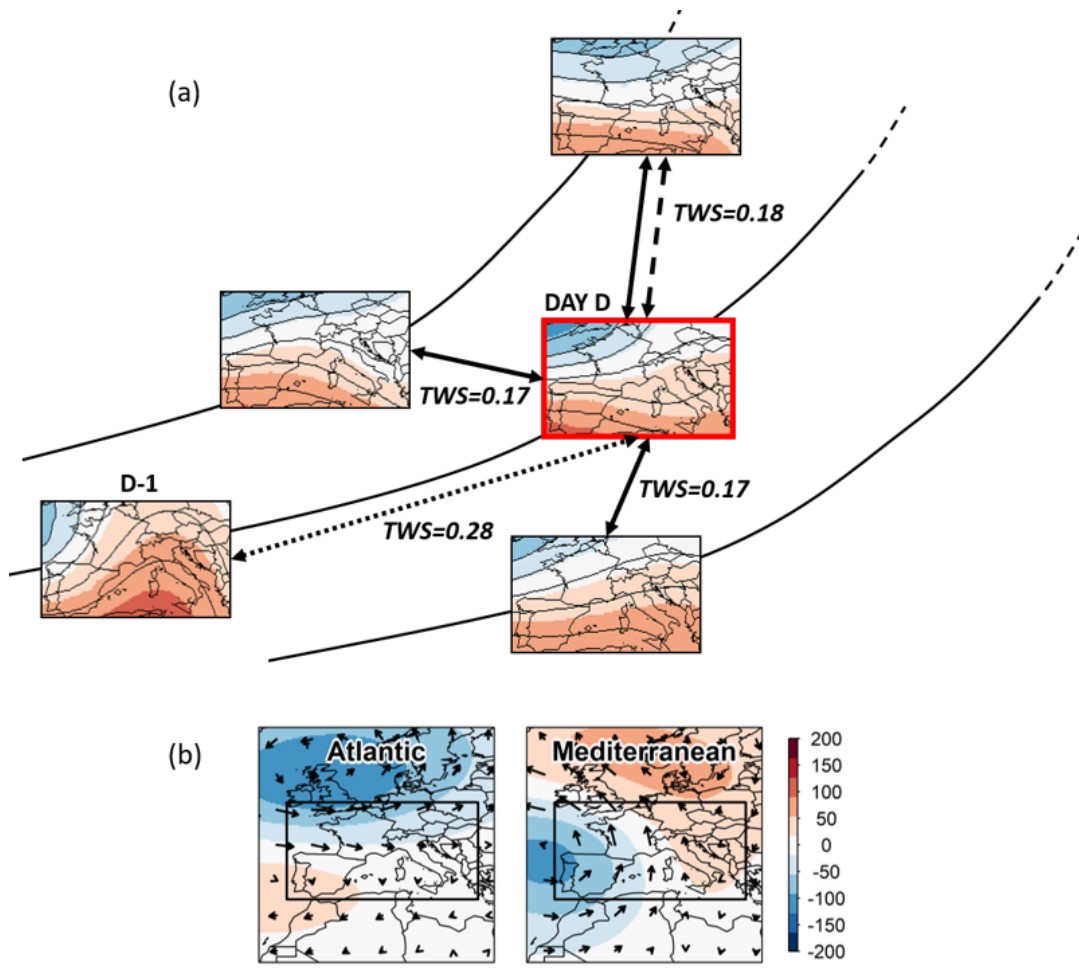

**Figure 2.** (a) Schematic illustration of the atmospheric descriptors based on analogy. Each map represents the 500 hPa geopotential height field over Western Europe for a given day. Black lines represent trajectories in phase space. Following days are represented on the same trajectory, but all trajectories are part of a single historical trajectory. The distance between each map represents here the difference in geopotential shapes/flow direction between individual days, using the Teweles-Wobus score. The scores are represented for indication. The celerity is the distance between a day D and the day before D-1 (dotted arrow). The singularity is the mean distance between a day D and its closest analogs (solid arrows; three analog days for illustration, but 111 analog days in our study). The relative singularity is the singularity normalized by the distance to the farthest analog (dashed arrow; third analog day for illustration, but 111th analog day in our study). The day D considered here is 12 December 1978. (b) Composite 500 hPa geopotential height anomalies (in meters) of the Atlantic and Mediterranean influences for the period 1950-2019. The arrows represent the wind anomalies at 500 hPa. The black rectangle represents the domain considered for the 500 hPa geopotential height.

The celerity, singularity and relative singularity are based on the TWS which is a normalized score focusing on geopotential shapes (that is on flow direction), whatever the range of geopotential heights governing the strength of the flow. Therefore we complement the three above analogy descriptors with the Maximum Pressure Difference (MPD) as fourth descriptor. The MPD of day $t_k$ is defined as the range of geopotential heights (in meters) over all grid points $j$ in Western Europe:

$$MPD_k = \max_j(z_{jk}) - \min_j(z_{jk}).$$ (5)

The higher $MPD_k$, the larger the pressure difference between the low and the high pressure systems at day $t_k$, i.e. the more pronounced the centers of action over Western Europe for this day. Although it reflects a pressure difference, the MPD over Western Europe appears to be weakly related to NAO (Blanc et al., 2021).

The four descriptors are computed for each day over the period 1851-2010 for 20CRv2c, 1900-2010 for ERA20C and 1950-2019 for ERA5. The results of the present paper are expressed per season, based on the daily time series of descriptors. The four seasons are defined as December-January-February (winter), March-April-May (spring), June-July-August (summer), and September-October-November (autumn).

## 3.2 Main atmospheric influences

We use the weather pattern classification of Garavaglia et al. (2010) from 1950 to 2019 to derive the main atmospheric influences driving precipitation in the Northern French Alps. This classification into eight weather patterns was established to link daily rainfall field shapes over Southern France with synoptic situations. The classification is based on a hierarchical ascendant classification of rainfall fields shapes over France. Non rainy days are then attributed to the weather pattern according to the proximity in 700 hPa and 1000 hPa geopotential shapes to the centroïds of the classes. Here, we aggregate the eight weather patterns into four atmospheric influences according to the origin of the air flow reaching the French Alps, as previously done in Blanc et al. (2021) and Blanchet et al. (2021b). We end up with four atmospheric influences: Atlantic circulations, Mediterranean circulations, Northeast circulations and Anticyclonic conditions. The Atlantic, Mediterranean, Northeast and Anticyclonic influences respectively account for 37 %, 25 %, 9 % and 29 % of days between 1950 and 2019. Trends in occurrence of the main atmospheric influences are small over this period, although a decreasing trend of the Northeast circulation is observed in spring and summer (Fig. 3 in Blanchet et al., 2021b). In this paper, we will only consider the Atlantic and the Mediterranean influences that are the two main influences driving precipitation in the Northern French Alps (Blanchet et al., 2021b; Sodemann and Zubler, 2010). Atlantic circulations correspond to an enhanced zonality of the flow while Mediterranean circulations correspond to low pressure anomalies over the near Atlantic, increasing the meridional component of the flow (Fig. 2b). Note that these two regional atmospheric influences are weakly related to NAO, although the occurrence of Mediterranean circulations in winter is slightly negatively correlated with NAO (Pearson correlation of $-0.45$, Blanc et al., 2021).

## 4 Results and Discussion

### 4.1 Past evolution of Western Europe LSC from 1851 to 2010 at seasonal scale

The atmospheric descriptors are first employed to study the long-term evolution of Western Europe LSC over the period 1851-2010. As most of the descriptors rely on analogy in geopotential shapes, we start the analysis by checking whether the different reanalyses provide similar geopotential shapes over this period. Fig. 3 shows the evolution of the Teweles Wobus Score (TWS) between the daily geopotential height fields obtained from different reanalyses. As an example, we compute the TWS in 500 hPa geopotential height fields between the $01/01/1900$ of the first member of 20CRv2c and the $01/01/1900$ of ERA20C, and we do the same for every day in the common period of the two reanalyses. We show in Fig. 3 the seasonal mean TWS of every year. ERA20C and ERA5 have been systematically interpolated on a coarser horizontal grid using a bilinear interpolation to allow for the computation of crossed TWS between reanalyses. 20CRv2c grid is used to compare 20CRv2c, ERA20C and ERA5; ERA20C grid is used to compare ERA20C and ERA5. Recalling that a TWS score of 0 represents two identical geopotential shapes, we observe that differences in geopotential shapes between reanalyses are weaker after 1950 than before (20CRv2c, ERA20C) and that differences remain quite steady from 1950 to 2010 (20CRv2c, ERA20C, ERA5). Before 1950, larger differences are observed between 20CRv2c and ERA20C, especially at the beginning of the 20th century. Those differences in geopotential shapes are larger in summer and weaker in winter while spring and autumn feature a transitional behavior. The fact that geopotential shapes are more defined in winter than in summer (larger MPD, see Fig. 7 of Blanc et al., 2022) makes it probably easier to capture the main pattern of the circulation even with few assimilated observations. Note however that the larger differences in summer geopotential shapes do not necessarily reflect larger differences in absolute air flow. As a reference, we add in Fig. 3 the TWS between day D and two other days considered in Fig. 2a that are equal to 0.18 and 0.28. Differences in shape between ERA20C and 20CRv2c (in red) before 1950 are notable; they are close to a TWS of 0.28, which reflects significant differences in geopotential shapes (see the difference in geopotential shapes between days D and D-1 in Fig. 2a). After 1950 and except in summer, differences in geopotential shapes between reanalyses are lower than a TWS of 0.18, which reflects quite close geopotential shapes (see Fig. 2a). Substantial differences in geopotential shapes are also observed between 20CRv2c members (in gray) from 1851 to 1880 but they always remain less pronounced than differences between 20CRv2c and ERA20C from 1900 to 1950. This is no surprise as they correspond to differences within the same reanalysis. Furthermore, it is interesting to note the larger differences between reanalyses and members during both World Wars due to the weaker number of assimilated observations. Overall, the significant differences in geopotential shapes before 1950 combined with the non-stationarity of the differences along the 20th century may have implications on the long-term evolution of LSC obtained from 20CRv2c and ERA20C.

In order to better understand the differences in shape between 20CRv2c and ERA20C, we map in Fig. 4 the differences in 500 hPa geopotential height between the first member of 20CRv2c and ERA20C for both periods 1900-1930 and 1970-2000 (first and second rows). Note that similar patterns are obtained using the second member of 20CRv2c (not shown). Over the period 1900-1930, 20CRv2c features a larger meridional pressure gradient than ERA20C, with larger 500 hPa geopotential height over Southern and Eastern Europe but lower 500 hPa geopotential height over Northwestern Europe, especially in winter.

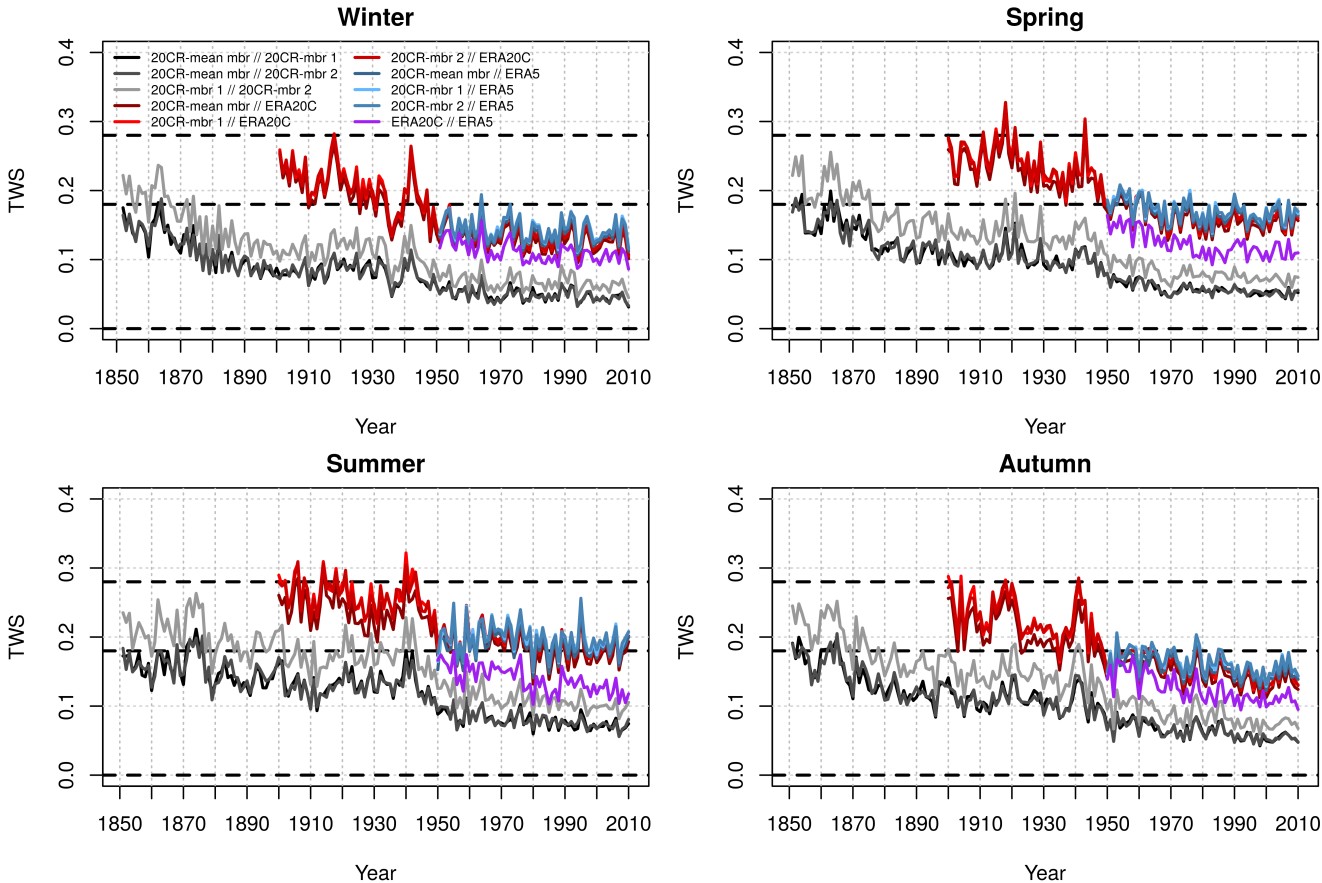

**Figure 3.** Evolution of the seasonal mean Teweles-Wobus Score (TWS) between the geopotential height fields from the different reanalyses over the period 1851-2010. The black dotted lines show respectively the minimum TWS value (0) – when geopotential shapes are identical – and, for reference, the TWS between day D and two other days in Fig. 2a – $TWS = 0.18$, and $TWS = 0.28$ which corresponds to the 69 % percentile of celerity for the period 1950-2010 using ERA5.

This pattern is reversed over the period 1970-2000, with 20CRv2c featuring larger 500 hPa geopotential height in Northern Europe than ERA20C. Overall, we can note that the shape of the geopotential height differences over Western Europe are more
pronounced over the period 1900-1930 than over 1970-2000, and they are also more pronounced in summer than in the other seasons, in accordance with Fig. 3. In terms of evolution, 20CRv2c shows mainly increases in the 500 hPa geopotential height between the two periods, with a more pronounced increase over Great Britain (Fig. 4, third row). ERA20C features an increase in the 500 hPa geopotential height mainly in Southern Europe while a decrease in observed in Northern and Northeastern Europe for almost every season (Fig. 4, fourth row). This decrease in 500 hPa geopotential height is located further North
from Western Europe in winter (black rectangle). The increase in the meridional pressure gradient in ERA20C between the beginning and the end of the 20th century is in line with Bloomfield et al. (2018) who show an increase in the Arctic Oscillation from October to March in ERA20C. This increase is not observed in two other observation products; it appears to be explained in ERA20C by a larger sea-level pressure in the North Pole in 1900 that decreases along the 20th century, while no trend is observed over Northern Europe (Fig. 4 of Bloomfield et al., 2018). The latter is not necessarily in contradiction with our
results since Bloomfield et al. (2018) study sea-level pressure while we study the 500 hPa geopotential height. At higher levels, this increase in meridional pressure gradient is consistent with Ménégoz et al. (2020) who show an increase in the westerly component of moisture flux over the Northern half of Europe using a regional climate model forced by ERA20C from 1902 to 2010 (Fig. 5 therein). It is also in line with Rohrer et al. (2019) showing an increasing storm track activity in ERA20C along the 20th century over the North Atlantic/European domain. Fig. 4 therefore highlights that the spatial differences in geopotential
shapes between ERA20 and 20CRv2c come out as a reinforcement of the meridional pressure gradient between 1900-1930 and 1970-2000 in ERA20C, that is not observed in 20CRv2c. Note however that this result does not allow for the selection of the "true" reanalysis, as the use of an independent data set would be required to inform such a choice.

Finally, we plot the evolution of the celerity, singularity, relative singularity and MPD over the period 1851-2010, considering a 5-year running average (Fig. 5). Overall, we observe a large interdecadal variability, especially for the singularity and MPD.
This variability is broadly similar between the different reanalyses over the whole period. Except in summer, ERA5 produces larger values of celerity, singularity and relative singularity in comparison to the long-term reanalyses, as well as larger MPD values in every season (colored dots in 2010, Fig. 5). This result is consistent with Rohrer et al. (2018) who show that high-resolution reanalyses tend to produce larger cyclone intensities and higher cyclone center densities, while full-input reanalyses tend to produce more intense blocking. The higher spatial resolution of ERA5 as well as the assimilation of surface, upper-air
and satellite observations generate more detailed geopotential shapes at 500 hPa, giving larger pressure differences (MPD) and weaker resemblances (celerity, singularity and relative singularity).

Over the period 1900-1950, major differences in descriptors trends are found between 20CRv2c and ERA20C. Differences are larger in summer and weaker in winter, as already observed for differences in geopotential shapes (Fig. 3). This result is in line with Rohrer et al. (2019), who show larger differences between 20CRv2c and ERA20C in summer than in winter
regarding trends in the 500 hPa geopotential height variability over the North Atlantic/European domain (Fig. 4 therein). It is however important to remind that the 500 hPa geopotential height variability and range are weaker in summer over this region (Rohrer et al., 2019; Blanc et al., 2022). The differences in descriptor trends remain considerable as they are clearly out of

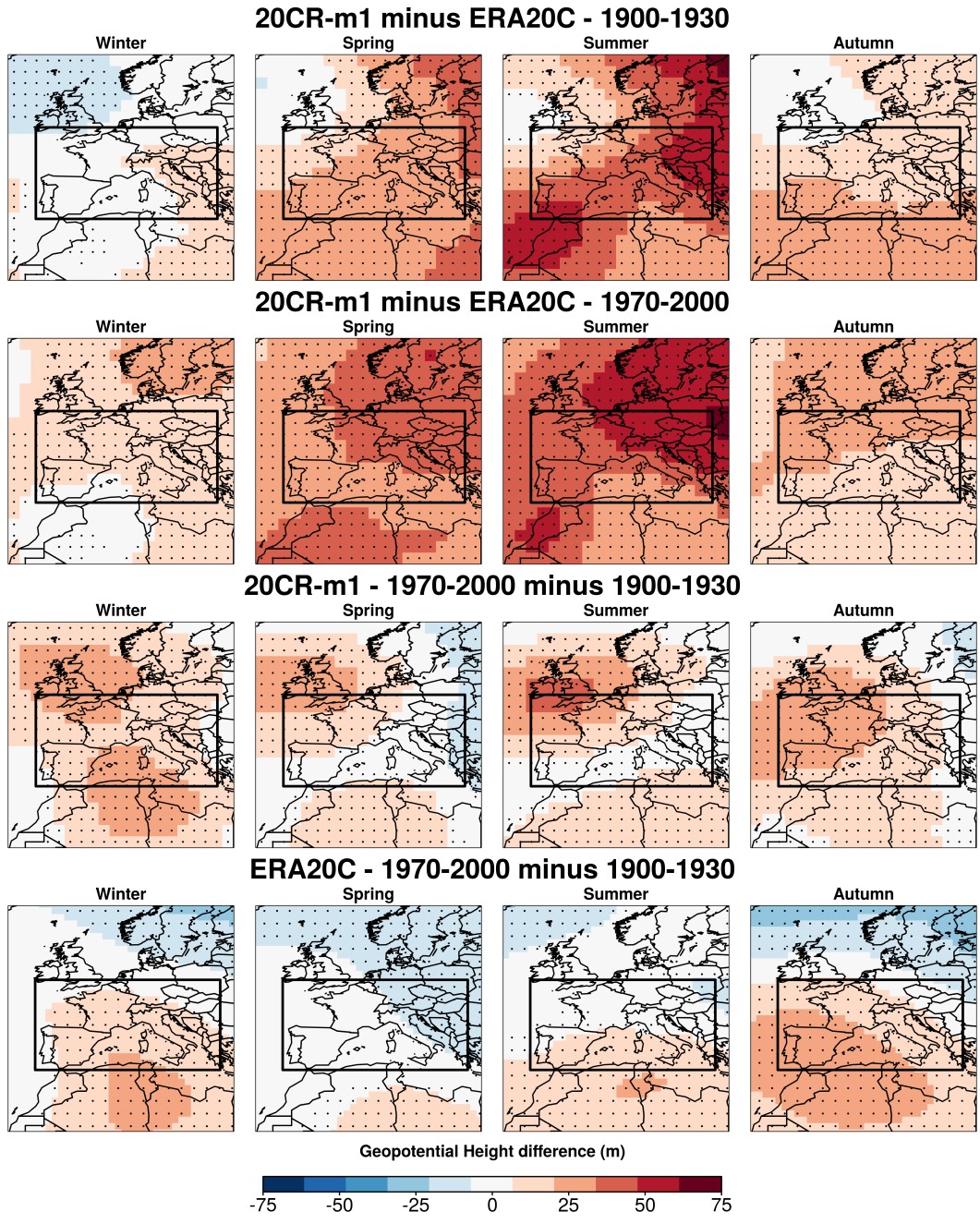

**Figure 4.** 500 hPa geopotential height difference (in meters) between the composites of i) the first member of 20CRv2c and ERA20C over the period 1900-1930 (first row), ii) the first member of 20CRv2c and ERA20C over the period 1970-2000 (second row), iii) the first member of 20CRv2c over the period 1970-2000 and 1900-1930 (third row), and iv) ERA20C over the period 1970-2000 and 1900-1930 (fourth row), per season. The dots indicate significant differences between the two periods according to a two sample t-test (pvalue<5 %). The Western Europe region over which the atmospheric descriptors are computed is represented by the black rectangle.

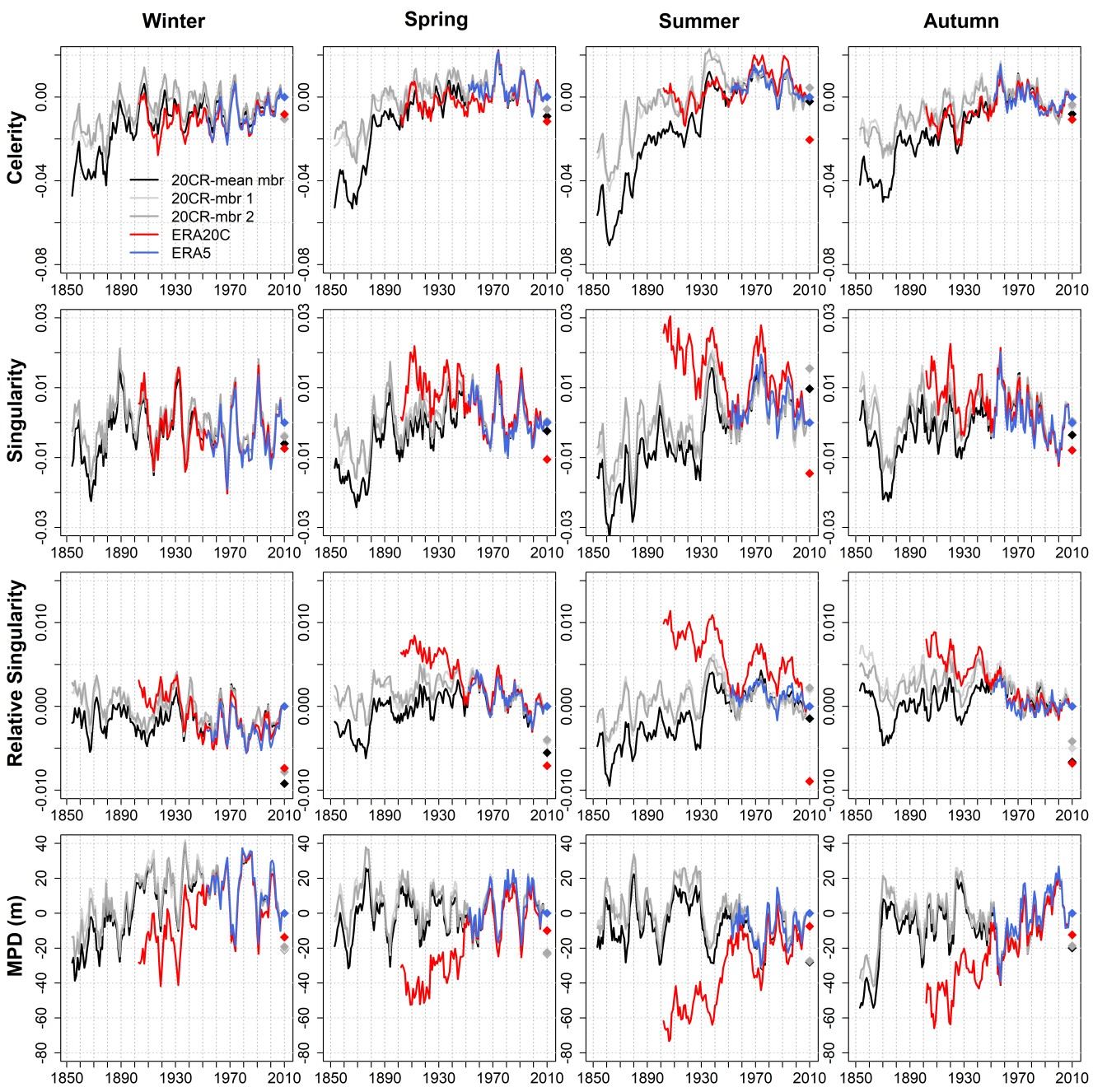

**Figure 5.** Evolution of the celerity, singularity, relative singularity and MPD per season over the period 1851-2010 for different reanalyses. A running average of 5 years is applied to allow a clearer visualisation. Time series are represented as anomalies according to their respective 2006-2010 average. The colored dots in 2010 indicate the differences in 2006-2010 average between ERA5 and the other reanalyses.

the range of the descriptor natural variability. Differences are more pronounced for the MPD, the relative singularity, and the singularity than for the celerity. ERA20C shows a strong trend at having more closely reproduced flow directions from 1900 to 1950, especially from spring to autumn (decreasing singularity and relative singularity). As the singularity of Western Europe geopotential shapes is related to the zonality of the flow (Blanc et al., 2022), this suggests a strengthening of the zonality of the flow from 1900 to 1950 in ERA20C. ERA20C also shows a strong trend at having more marked geopotential shapes from 1900 to 1950 (increasing MPD), in accordance with a reinforcement of the meridional pressure gradient (Fig. 4). These major differences in LSC trends between ERA20C and 20CRv2c are in line with several studies showing inconsistencies in wind speed trends between the two reanalyses before 1950 (Befort et al., 2016; Wohland et al., 2019). The assimilation of marine wind and the increasing number of associated observations in ERA20C is pointed out as the main driver of the increasing wind speed in the reanalysis in the first half of the 20th century – a trend that is neither observed in the model-only integration ERA20CM nor in 20CRv2c which only assimilates surface pressure (Meucci et al., 2019; Wohland et al., 2019). Trends in wind speed may have impacted pressure at both sea level (Bloomfield et al., 2018) and higher elevations.

Substantial differences in trends are also found between the individual members and the mean member of 20CRv2c before 1950. Differences are also more pronounced from spring to autumn, and they are quite pronounced for the celerity. The mean member of 20CRv2c shows lower values of celerity from 1850 to 1880 together with a strong increase in celerity from 1880 to 1950, suggesting more stationary flow directions in the 19th century. This feature is however much less pronounced in 20CRv2c individual members. Notable differences between 20CRv2c mean and individual members are also observed before 1950 regarding the singularity and relative singularity. The reduced quantity of assimilated data in the beginning of the 20CR reanalysis (see Fig. 2 of Wang et al., 2012) could explain i) the generation of smoother individual members, which allows for closer analogs and explains the systematically lower celerity and singularity in the beginning of the reanalysis, and ii) the larger differences in geopotential shapes between individual members, leading to a smoothed mean member and lower celerity, singularity and relative singularity in comparison to individual members. This is consistent with the study of Brönnimann et al. (2012) focusing on extreme wind trends in 20CRv2 and with the study of Rodrigues et al. (2018) focusing on trends in the dynamical properties of the North Atlantic circulation in 20CRv2c, which both point out the ensemble mean as more suitable than the mean member to derive long-term trends. This is reinforced by the fact that the two individual members mostly share the same evolution in LSC characteristics even with quite different geopotential shapes (Fig. 3). Furthermore, it is interesting to note that differences in MPD between individual and mean member are rather weak over the whole period, meaning that averaging individual members leads to smoother but not flatter geopotential shapes. This reflects that the location and the intensity of the centers of action in individual members of 20CRv2c are similar over the whole period of the reanalysis, while the other regions of the pressure fields are less constrained, and are thus more variable in shape. As previously exposed, the fact that geopotential shapes are more marked in winter (larger MPD, see Fig. 7 of Blanc et al., 2022) makes it easier to capture the main pattern of the circulation even with few assimilated observations. This leads to weaker differences in geopotential shapes between individual and mean member in winter before 1950 (Fig. 3). Overall, the lower number of assimilated observations in the beginning of the 20CR reanalysis and the differences between individual and mean members suggest that the increasing celerity and singularity in the second half of the 19th century is more an artefact of the data set than a true physical signal.

From 1950 to 2010, there is a good agreement between the different reanalyses. This result is in line with the weaker differences in geopotential shapes observed between reanalyses after 1950. We can note a negative trend in relative singularity in spring from 1950 to 2010, which is consistent with the decreasing local dimension of Rodrigues et al. (2018) and Faranda et al. (2019) over the North Atlantic, pointing out to a decrease in the number of degrees of freedom around the atmospheric states. We can also note the increase in MPD in autumn, pointing out to an increasing intensity of the centers of action over Western Europe from September to November.

To summarize, the interannual and interdecadal LSC variability is consistent between the three reanalyses and highlights the large natural variability affecting Western Europe LSC. However, substantial differences in LSC trends are observed between reanalysis before 1950, making the physical interpretation of the trends difficult. ERA20C features less marked and quite different geopotential shapes in comparison to 20CRv2c in the early 20th century, as well as a clear increase of the meridional pressure gradient until 1950. This result is consistent with the literature which shows that the pronounced trends in ERA20C might be driven by the increasing trend in the assimilated marine wind – a variable that is not assimilated in 20CRv2c. Furthermore, significant differences are also found between the geopotential shapes of 20CRv2c members before 1950, which is probably related to the low number of assimilated data in the beginning of the reanalysis. The large differences in Western Europe LSC between long-term reanalyses hence make the study of LSC evolution difficult before 1950. This also suggests that LSC trends obtained before 1950 should be taken with caution when using these products. In order to look in more details on the trends in LSC characteristics after 1950, we now focus on the distribution of daily descriptors instead of their seasonal mean and we distinguish the main atmospheric influences driving precipitation in the Northern French Alps.

## 4.2 Recent changes in Western Europe LSC from 1950 to 2019 at daily scale

We focus on the changes in Western Europe LSC from 1950 to 2019 according to ERA5. We take advantage of the atmospheric descriptors to study changes in the whole descriptor distribution at the daily scale, rather than only considering trends in mean descriptor values over a season as we did in Section 4.1. To do this, we separate the period 1950-2019 into two sub-periods of 35 years each and we look at the changes in descriptor distribution between the two sub-periods. We consider separately the days associated with either the Atlantic and or the Mediterranean influence to detect changes that are specific to the given influence. Both Kolmogorov-Smirnoff and Anderson-Darling tests are carried out to detect significant differences in descriptor distribution at 5 % level. The significant differences in descriptor distribution and the sign of the difference in average descriptor value between the two sub-periods are summarized in Table 2. We observe that the differences in descriptor distribution are not equally distributed between the two influences (Table 2). Atlantic circulations show few significant differences while Mediterranean circulations show significant differences in every season but especially in summer and autumn. In the following, the LSC changes of Table 2 are studied in more details.

### 4.2.1 Atlantic circulations

Fig. 6 shows the descriptor distribution of Atlantic circulations (boxplots) as well as the differences in descriptor densities between 1985-2019 (referred as the present period) and 1950-1984 (referred as the early period), per season. Over the present

| | Winter | | | | Spring | | | | Summer | | | | Autumn | | | |
| --- | --- | --- | --- | --- | --- | --- | --- | --- | --- | --- | --- | --- | --- | --- | --- | --- |
| | cel | sing | rsing | MPD | cel | sing | rsing | MPD | cel | sing | rsing | MPD | cel | sing | rsing | MPD |
| Atlantic | | - | | | | | - - | | - - | | - | | | | | + |
| Mediterranean | | + + | | - - | + | + + | | - | - - | - - | | + + | - | - - | | + + |

**Table 2.** Significant differences in descriptor distribution between the period 1985-2019 and the period 1950-1984 for Atlantic and Mediterranean circulations. Differences are considered significant if the p-value of either the Kolmogorov-Smirnoff test or the Anderson-Darling test is lower than 5 %. Differences that are significant with both tests are marked with a double sign. The sign indicates whether the average descriptor value has increased (+) or decreased (-) from 1950-1984 to 1985-2019.

period, Atlantic circulations feature a decreasing celerity in summer, which suggests an increasing stationarity of zonal flows. Atlantic circulations feature a decreasing relative singularity in spring and to a lesser extent in summer, pointing out to more Atlantic circulations featuring a high degree of clustering with their closest flow directions. Finally, over the present period, Atlantic circulation feature slightly more pronounced centers of action in autumn (increasing MPD) and more closely reproduced flow directions in winter (decreasing singularity), with enhanced reproducibility particularly for the most closely reproduced flows (below the 25 % percentile of singularity). This result is consistent with Yiou et al. (2018) who show that winter circulations over the North Atlantic tend to become more similar to already known patterns, with the dominant atmospheric patterns – mainly NAO+/zonal patterns – being trapped for longer times within the winter season. Looking at the spatial patterns of the differences, we observe that changes in 500 hPa height are quite weak from spring to autumn (Fig. 7). Winter definitely shows the largest differences with a marked increase in 500 hPa heights over Northern Italy and a decrease in the Northwest of Great Britain. According to the anomalies associated with Atlantic circulations (Fig. 2b), this pattern reflects i) a northward shift of the Atlantic storm track between the two sub-periods, and ii) an increasing southwest component of Atlantic circulations. The latter is consistent with a decreasing singularity, the least singular geopotential shapes for Western Europe featuring west-to-southwest flow directions (Fig. 6 in Blanc et al., 2022).

### 4.2.2 Mediterranean circulations

Mediterranean circulations feature large differences in descriptor distribution between the early and the present period with opposite differences across the seasons (Fig. 8). In summer and autumn, Mediterranean circulations become more stationary (lower celerity), more marked (larger MPD) and more reproducible in shape. In autumn, this shift in LSC characteristics corresponds to more than a doubling (from 0.7 % to 1.7 %) in the proportion of Mediterranean circulations featuring among the most stationary, the most closely reproduced (< 10 % percentile of celerity, singularity, relative singularity) and the most pronounced geopotential shapes (> 90 % percentile of MPD), and still a 30 % increase (from 7.9 % to 10.4 %) in the proportion of Mediterranean circulations featuring quite stationary, closely reproduced (< 30 % percentile of celerity, singularity, relative singularity) and pronounced geopotential shapes (> 70 % percentile of MPD). In winter and spring, Mediterranean circulations tend to become more singular and less marked as well as less stationary (Fig. 8). The seasonal contrast of the differences in

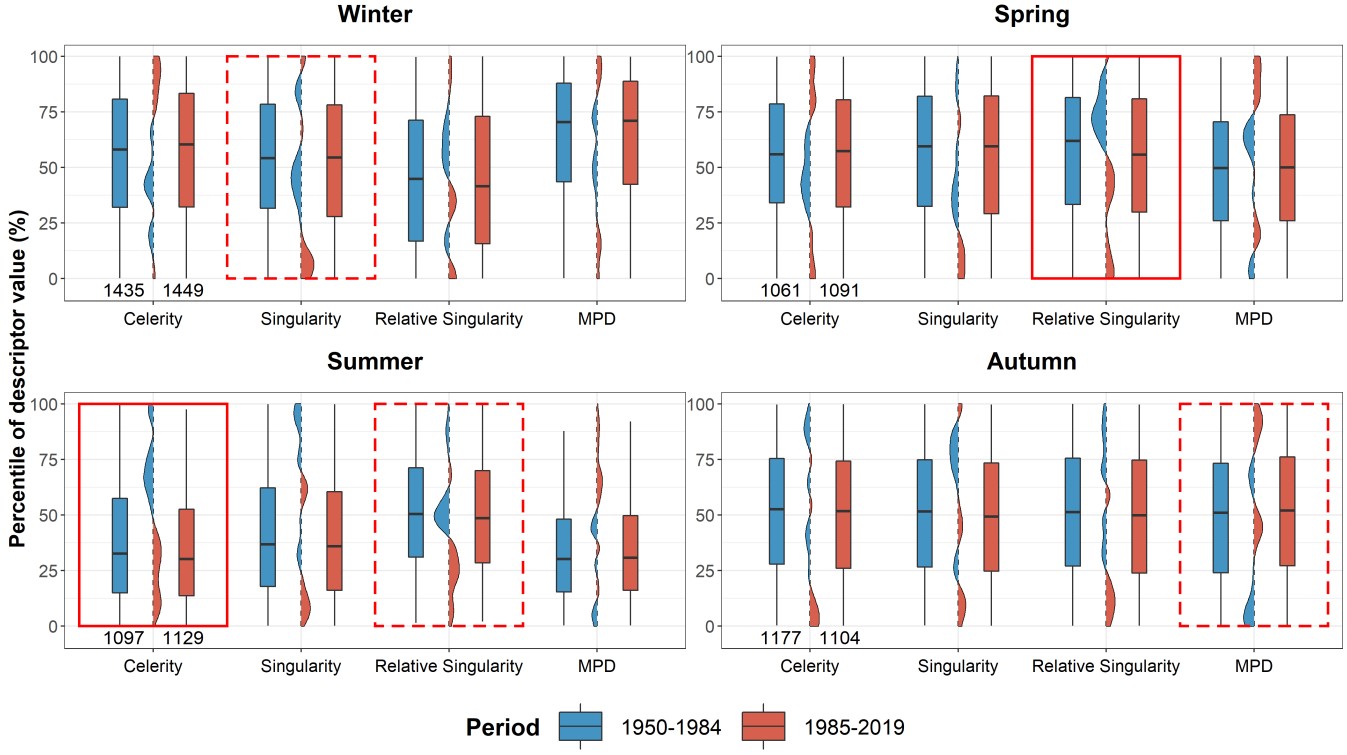

**Figure 6.** Boxplot of the daily celerity, singularity, relative singularity and MPD of Atlantic circulations for the two 35-year periods of 1950-1984 and 1985-2019, per season. Descriptor values are represented as percentiles to allow the representation of the four descriptors on the same axis. Percentiles are computed with respect to all days of 1950-2019 belonging to Anticyclonic conditions. The difference in density between the two sub-periods is shown between the boxplots. The range of the density that is colored in blue (respectively red) means that the considered descriptor shows more values within this range in the early (respectively present) period. The numbers in the bottom left of the graphs indicate the number of days considered in the early and present periods. A continuous red rectangle indicates the descriptor and season where the difference in distribution between the two sub-periods is significant according to both the Kolmogorov-Smirnov test and the Anderson-Darling test. A dashed red rectangle indicates significance with only one of the two tests. The difference is considered significant if the p-value of the test is lower than 5 %.

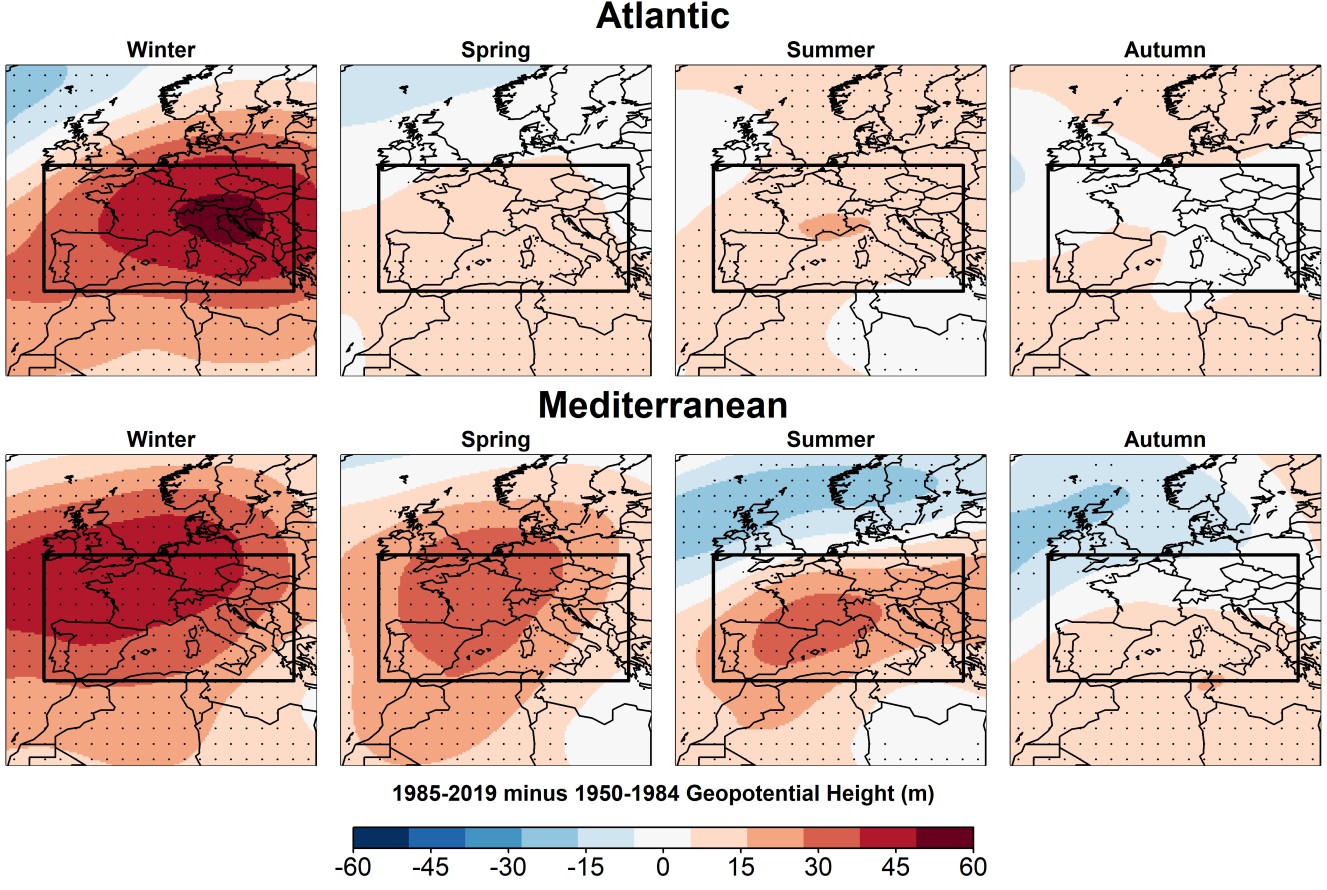

**Figure 7.** 500 hPa geopotential height difference (in meters) between 1985-2019 and 1950-1984 according to the ERA5 reanalysis for Atlantic circulations and Mediterranean circulations, per season. The dots indicate significant differences between the two periods according to a two sample t-test (pvalue<5 %). The Western Europe region over which the atmospheric descriptors are computed is represented by the black rectangle. The maximum geopotential height difference displayed here reaches 60 m. This correspond to 30 % of the maximum anomalies of 200 m associated with the main atmospheric influences in Fig. 2b.

Mediterranean circulations is clearly visible in the maps of Fig. 7. Reminding that Mediterranean circulations are associated to low pressure anomalies over the near Atlantic (Fig. 2a), the large increase in 500 hPa geopotential height over the whole Northwestern Europe region in winter and to a lesser extent in spring confirms the weakening of Mediterranean circulations over the present period during these seasons. In summer and autumn, a decrease in 500 hPa geopotential height is observed over Northwestern Europe, suggesting low pressure anomalies located further North over the present period. This is fully consistent

with the trends in summer 500 hPa circulation patterns over Europe for the period 1979-2013, showing an increasing occurrence of low pressure anomalies over the near Atlantic close to Ireland and a decreasing occurrence of low pressure anomalies centered over Northern Portugal (Extended Data Fig. 2 of Horton et al., 2015). The decrease in 500 hPa geopotential height over Northwestern Europe reaches further South in autumn and it is associated with a slight increase in 500 hPa geopotential height over Southern Europe. This reflects a reinforcement and an increasing zonality of autumn Mediterranean circulations,

suggesting more frequent Southwestern flows at 500 hPa. Blanc et al. (2022) have shown that the singularity of Western Europe LSC is related to the zonality of the flow – the more zonal circulations being the more closely reproduced in the climatology. This applies here for autumn Mediterranean circulations that became more zonal and less singular. Summer Mediterranean circulations also became less singular, but the larger increase in 500 hPa geopotential height over Central Europe than over the near Atlantic close to Portugal reflects more a reinforcing of summer Mediterranean circulations than an increasing zonality.

This suggests that, in combination with the zonality of the flow, the singularity of Western Europe LSC is also related to the latitudinal position of the low pressure systems – a northern position being associated with lower singularities.

## 4.3 Link to precipitation trend in the Northern French Alps

We now discuss the link between these recent LSC changes and precipitation trend in the Northern French Alps. We focus on precipitation falling in the Isère River catchment and in the Drac River catchment at Grenoble over the period 1950-2017.

This period is two years shorter than in Section 4.2, but considering LSC changes over the period 1950-2017 leads to the same conclusions as Section 4.2 (not shown).

### 4.3.1 Seasonal precipitation

Fig. 9 shows the evolution of seasonal precipitation for the two catchments over the period 1950-2017 (black lines). The increasing spring precipitation in the Isère River catchment is the only significant trend. However, by considering separately

precipitation driven by Atlantic and Mediterranean circulations, we observe an increase in spring precipitation under Atlantic circulations and a decrease in winter precipitation under Mediterranean circulations in both catchments. The decreasing winter precipitation under Mediterranean circulations is more pronounced for the Drac River catchment that is more influenced by the Mediterranean Sea, with almost 50 % decrease in 68 years. Note that these results cannot be related to a change in the occurrence of Atlantic and Mediterranean circulations, as precipitation accumulation are normalized by the occurrence of the given

influence in each season. Anyway, the occurrence of Atlantic and Mediterranean circulations barely change over the period (Fig. 3 of Blanchet et al., 2021b), and similar trends are obtained considering absolute precipitation (not shown). This reflects an increasing contribution of Atlantic circulations to spring precipitation and a decreasing contribution of Mediterranean circu-

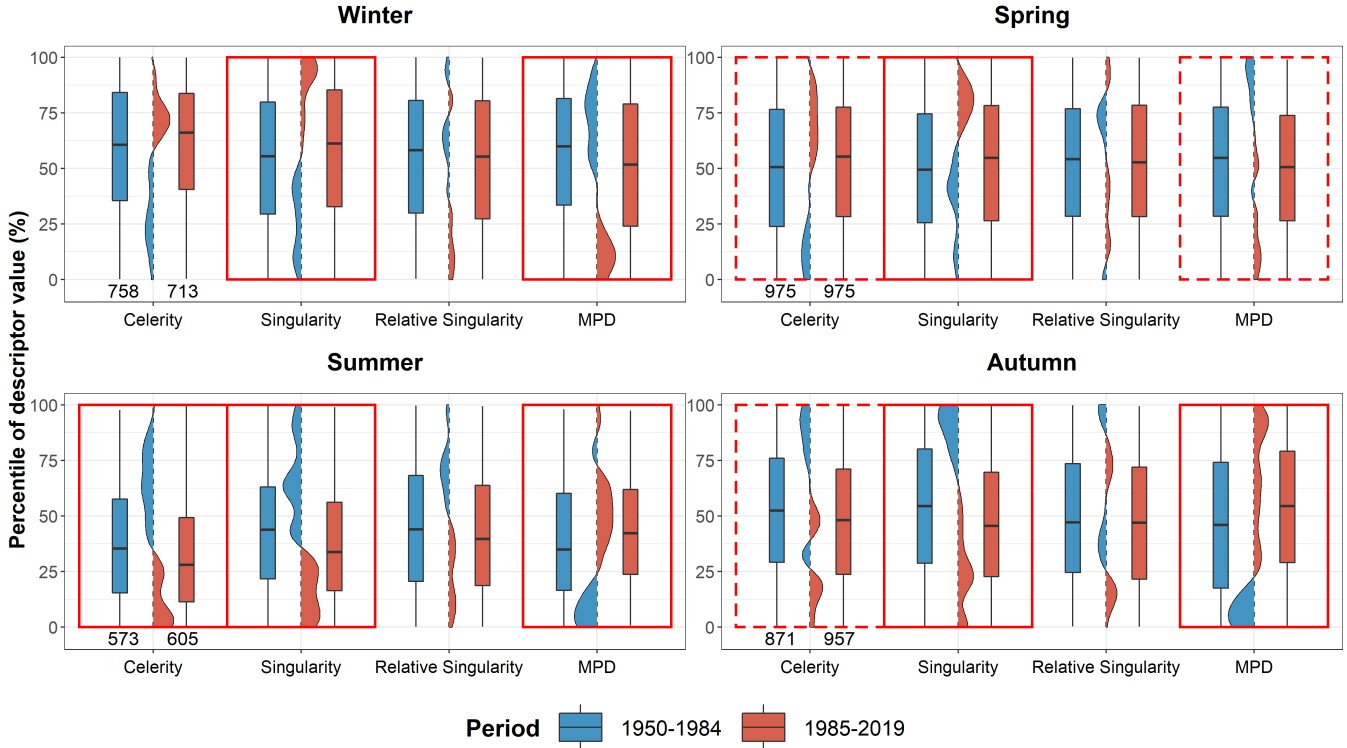

**Figure 8.** Same as Fig. 6, but for Mediterranean circulations.

lations to winter precipitation. We will further discuss the role of LSC changes for i) the increasing spring precipitation under Atlantic circulations in the Isère catchment and ii) the decreasing winter precipitation under Mediterranean circulations in the Drac catchment.

The increasing spring precipitation under Atlantic circulations is associated with little changes in LSC. Indeed, Atlantic circulations feature only a significant decrease in relative singularity, while every other descriptors show non-significant changes (Fig. 6). However the relative singularity is little explicative of "usual" precipitation, as shown in Fig. 4 of Blanchet and Creutin (2020) with CRPSS values. Another way of illustrating this is to consider the descriptor percentiles associated with wet (>1mm) and dry days in spring for the Isère River catchment, as shown in Fig. 10, left. Although the singularity and MPD appear quite relevant to discriminate wet and dry days – in accordance with Blanc et al. (2021), who show that these descriptors explain a significant part of precipitation variability in the Northern French Alps–, this is not the case for the celerity and relative singularity. Given the small changes in singularity and MPD (Fig. 6), it appears difficult to link the increasing spring precipitation under Atlantic circulations with a change in LSC. Studying trends in thermodynamic variables – such as the trend in humidity under spring Atlantic circulations – is an interesting perspective.

The decreasing winter precipitation under Mediterranean circulations is associated with significant changes in LSC. Winter Mediterranean circulations feature a clear shift towards more singular flow directions (larger singularity) and less pronounced

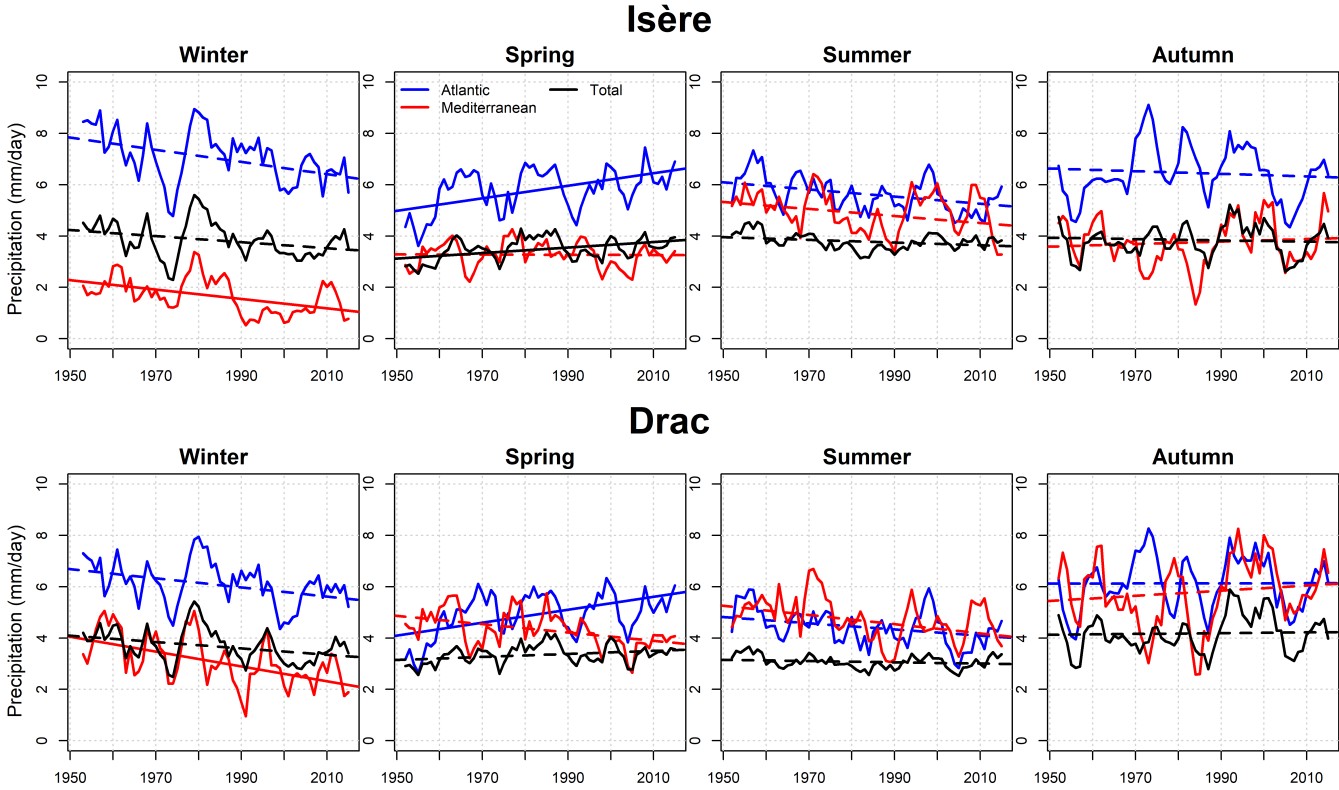

**Figure 9.** Evolution of seasonal precipitation (mm/day) for the Isère River catchment (top) and the Drac River catchment (bottom) over the period 1950-2017. For Atlantic and Mediterranean circulations, the seasonal precipitation is normalized by the number of days associated with the given influence. A running average of 5 years is applied to allow a clearer visualisation. Non-significant trends are represented by dotted lines. The trend is considered as significant if the p-value of the Student test is lower than 5 %.

centers of action (smaller MPD) over the period 1950-2019 (Fig. 8). These two descriptors appear relevant to discriminate wet and dry days in the Drac catchment, with wet days associated to larger MPD and lower singularity than dry days (Fig. 10, right). The decreasing occurrence of Mediterranean circulations featuring quite low singularities and quite large MPD – typically associated with Southwestern flows – in winter therefore imply less frequent wet days and hence smaller seasonal precipitation under Mediterranean circulations. This is consistent with the reversed LSC changes observed in autumn associated with an increase in precipitation under Mediterranean circulations in the Drac catchment (although non significant).

### 4.3.2 Extreme precipitation

We now focus on trends in daily seasonal maxima of precipitation independently of the atmospheric influence (Fig. 11), as the series of seasonal maxima associated to a given influence is impacted by the occurrence of this influence which varies from one year to another. The trends in mean maxima (blue lines) are obtained using the best non-stationary GEV (Generalized Extreme

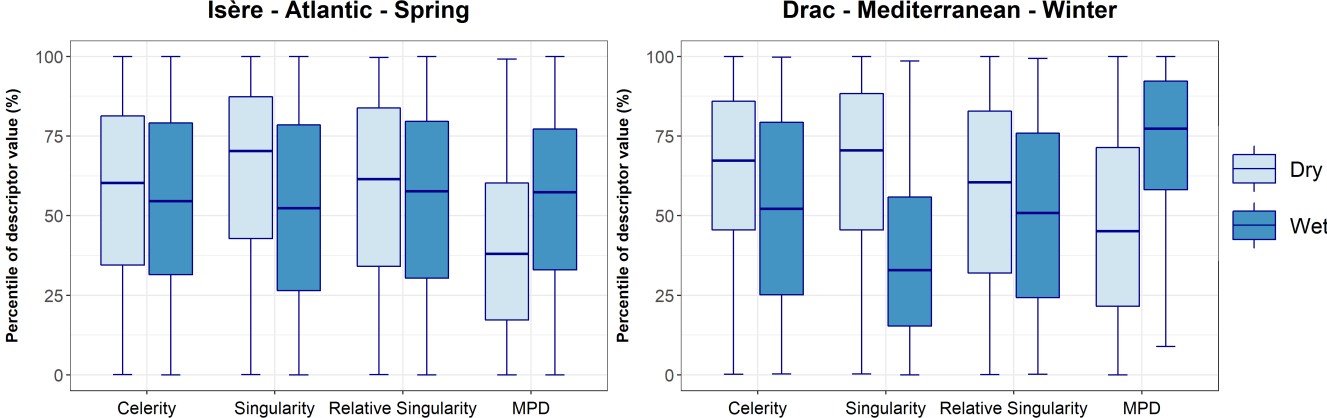

**Figure 10.** Percentile of descriptor value associated with dry days (light blue) of wet days (>1 mm, royalblue) for the Isère River catchment in spring under Atlantic circulations (left) and for the Drac River catchment in winter under Mediterranean circulations. Percentiles are computed with respect to the days associated to the given atmospheric influence over the period 1950–2017.

Value) model, following the methodology of Blanchet et al. (2021a). Such models can consider an evolution in both the mean value and the variability of the maxima. We observe a significant decrease in winter extreme precipitation in the Isère River catchment and a significant increase in autumn extreme precipitation in the Drac River catchment. The magnitude of the trends are quite small, but note that they are more pronounced for larger return levels (Blanchet et al., 2021a). In the following, we will consider these trends to discuss the potential links with LSC changes.

Extreme precipitation in the Isère River catchment in winter are driven at 94% by Atlantic circulations. Fig. 12 shows the percentile of descriptor values at the dates of winter precipitation maxima in the Isère River catchment, per atmospheric influence (left). Atlantic circulations driving extreme precipitation feature among the most pronounced centers of action (large MPD) with flow directions that are quite closely reproduced in the climatology (quite low singularity and relative singularity) in comparison to the other Atlantic circulations. The only significant evolution in winter Atlantic circulations is a decreasing singularity (Fig. 6), which suggests slightly more frequent Atlantic circulations likely to drive extreme precipitation. Therefore, the decreasing winter extreme precipitation in the Isère catchment does not appear to be associated with a decreasing occurrence of the driving LSC. Looking at the descriptor percentiles at the dates of the extremes (Fig. 13, left), we do not observe significant trend although a slight decrease in singularity and increase in MPD can be noted (note that the same evolution are obtained using absolute descriptor values, not shown). Considering LSC only, these slight trends would suggest an increase rather than a decrease in extreme precipitation (Blanchet et al., 2018; Blanc et al., 2022). Thus the decreasing winter extreme precipitation in the Isère River catchment cannot be explained by changes in LSC driving the extremes based on the present descriptors. Studying changes in temperature and humidity at the dates of winter extreme precipitation is an interesting perspective, but this is out of the scope of the present study.

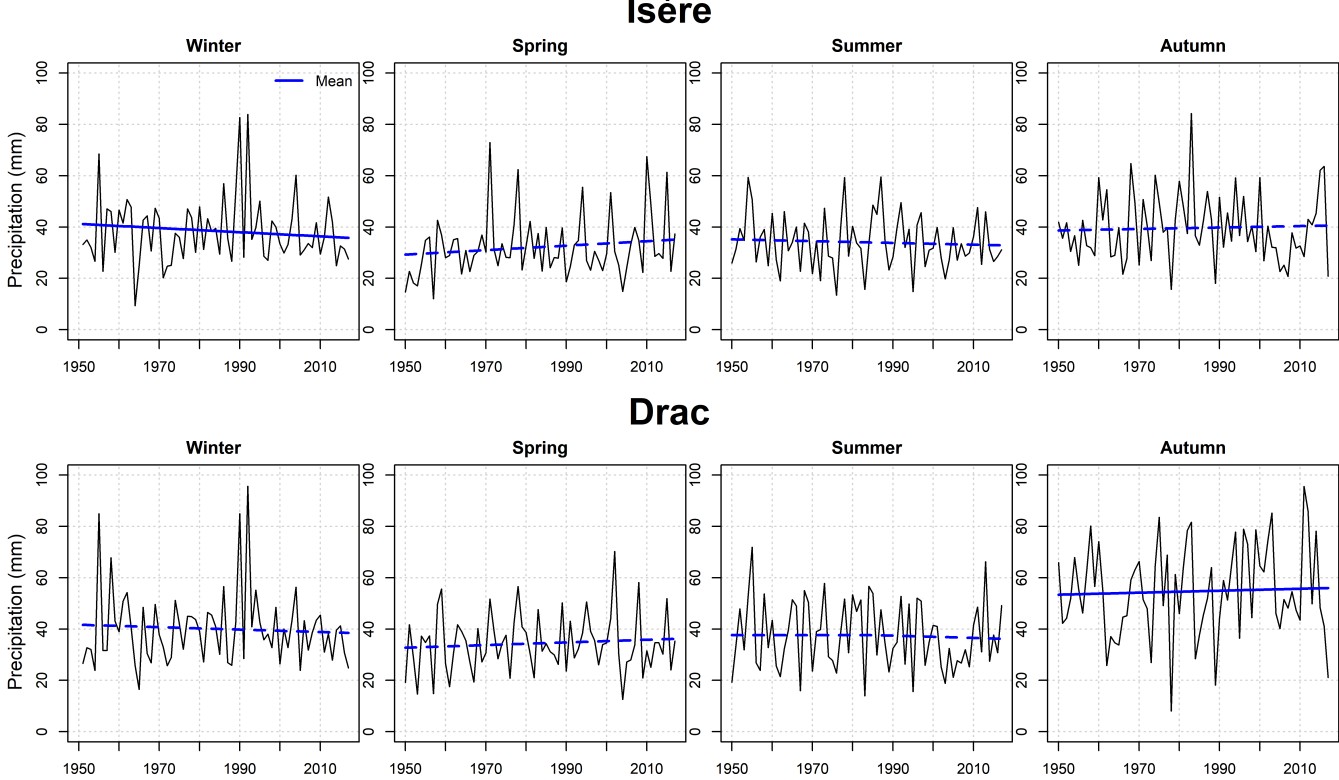

**Figure 11.** Seasonal maxima of daily precipitation for the Isère River catchment (top) and the Drac River catchment (bottom) over the period 1950-2017. The blue lines represent the evolution of the mean seasonal maximum obtained with the best non-stationary GEV model, following the methodology of Blanchet et al. (2021a). Non-significant trends are represented by dotted lines. The trend is considered as significant if the non-stationary model is significantly better than the stationary model according to the likelihood ratio test (pvalue<5 %).

Extreme precipitation in the Drac River catchment in autumn are driven at 53% by Mediterranean circulations and 44% by Atlantic circulations. LSC driving extreme precipitation feature among the most pronounced centers of action (large MPD) with flow directions that are quite closely reproduced in the climatology (quite low singularity and relative singularity) for their respective influence, especially for Mediterranean circulations (Fig. 12, right). The only significant evolution in autumn Atlantic circulations is an increasing MPD (Fig. 6), which suggests slightly more frequent Atlantic circulations likely to drive extreme precipitation. Furthermore, Mediterranean circulations being among the most pronounced and among the least singular – typically the strong Southwestern flows – appear to become more frequent (Fig. 8). This suggests that the increasing autumn extreme precipitation in the Drac catchment is associated with an increasing occurrence of the driving LSC, especially of the Mediterranean driving LSC. We do not observe significant trend in descriptor percentiles at the dates of the extremes (Fig. 13, right). LSC driving extreme precipitation even tend to feature less stationary (increasing celerity), less closely reproduced geopotential shapes (increasing singularity and relative singularity) and less pronounced centers of action (decreasing MDP) –

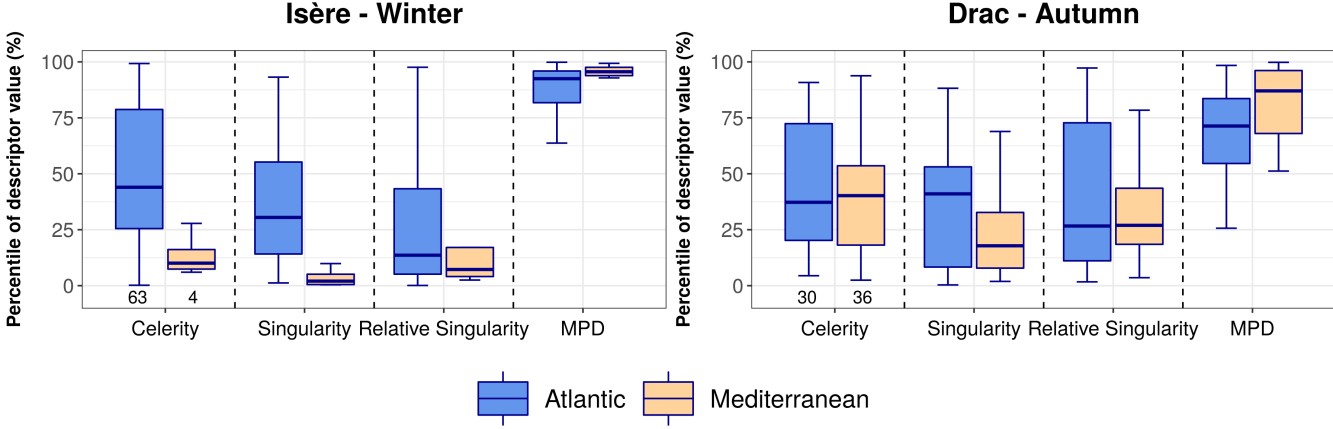

**Figure 12.** Percentile of descriptor value associated with the daily seasonal maxima of precipitation for the Isère River catchment in winter (left) and the Drac River catchment in autumn (right), per atmospheric influence. Percentiles are computed with respect to the days associated to the given atmospheric influence over the period 1950–2017. The numbers in the bottom left of the graphs indicate the number of days considered in the Atlantic and Mediterranean influences.

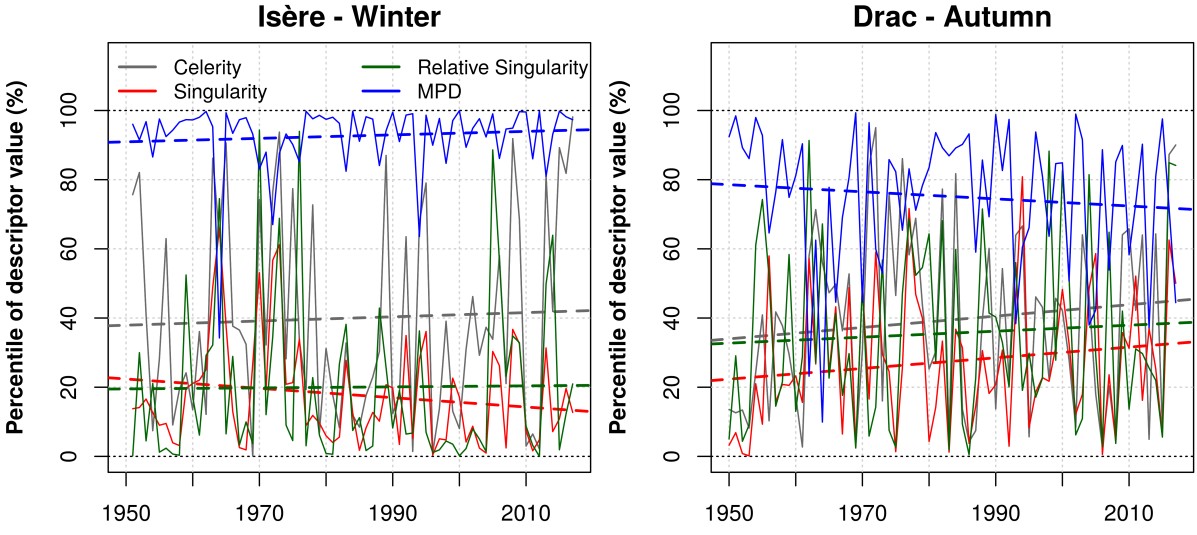

**Figure 13.** Evolution of descriptors percentiles at the date of the seasonal maxima of daily precipitation for the Isère River catchment in winter (left) and the Drac River catchment in autumn (right) over the period 1950-2017. Percentiles are computed with respect to all days of the period 1950-2017. Non-significant trends are represented by dotted lines. The trend is considered as significant if the p-value of the Student test is lower than 5 %.

which would suggest a decrease rather than an increase in extreme precipitation considering the atmospheric descriptors only. Therefore, although the increase in autumn extreme precipitation is partly driven by an increasing occurrence of the driving LSC, other variables such as humidity are also likely to play a role in explaining such trend. This is in line with the observed increase in extreme precipitation in the whole Mediterranean-influenced region of the Southwestern Alps in autumn (Blanchet et al., 2021a).

## 5    Conclusions

We have studied the past evolution of Western Europe large-scale circulation based on the 500 hPa geopotential height fields using different reanalyses products. We employed several atmospheric descriptors that are mostly based on analogy and that allow a quantitative characterization of daily LSC.

We first focused on large-scale circulation evolution from 1851 to 2010 at seasonal scale. We showed major trend differences before 1950 between 20CRv2c and ERA20C, in accordance with the literature. The two reanalyses feature quite different geopotential shapes in the first half of the 20th century, especially from spring to autumn. ERA20C produces flatter geopotential shapes in the beginning of the 20th century and an increase in the meridional pressure gradient that is not observed in 20CRv2c. In 20CRv2c, the lower number of observations that are assimilated in the second half of the 19th century could lead to the generation of smoother geopotential shapes and may be responsible for the differences in geopotential shapes between the individual members, especially between 1850 and 1880. Overall, the differences in geopotential shapes in long-term reanalyses highlight that LSC trends must be taken with caution before 1950.

We then focused on the changes in large-scale circulation after 1950 when the different reanalyses agree, according to ERA5. The atmospheric descriptors have been combined to an existing weather pattern classification to study large-scale circulation changes in the main atmospheric influences driving precipitation in the Northern French Alps. On the one hand, we have shown that winter Atlantic circulations tend to be shifted northward and they become more similar to known Atlantic circulations. On the other hand, we have shown that autumn Mediterranean circulations featuring a marked and stationary flow that is closely reproduced in the climatology – typically strong Southwestern flows towards the Northern French Alps – became more frequent over the last 30 years. In winter, opposite trends in Mediterranean circulations are observed.

Finally, we discussed the role of these recent changes in Western Europe large-scale circulation for precipitation in the Northern French Alps. This was done considering precipitation over the period 1950-2017 in two medium size mountainous catchments, namely the Isère and the Drac River catchments. The observed increase in spring precipitation in both catchments under Atlantic circulations do not appear to be related to a change in LSC. However, the decreasing winter precipitation in both catchments (and especially for the Drac catchment) under Mediterranean circulations seems to be driven by LSC changes – the Mediterranean circulations becoming less pronounced and less closely reproduced in winter over the last 30 years. Regarding extreme precipitation, the winter decrease in the Isère catchment do not appear to be driven by LSC changes, according to the descriptors used. However, the autumn increase in the Drac catchment is associated with an increasing occurrence of the driving Mediterranean circulations. Other variables, such as humidity, may also play a role for explaining this increase.

The present work faces some limitations. The 70-year period considered from 1950 to 2019 allows the detection of changes in large-scale circulations, but this is still too short to deduce long-term trends in large-scale circulation given the large natural variability. Moreover, the present study cannot assess whether the observed trends in large-scale circulation are a clear signal of climate change through anthropogenic forcing or a simple result of natural variability.

This article provides a view on the observed evolution of regional large-scale circulation over Western Europe, that is partly relevant to understand the evolution of precipitation in the Northern French Alps. This opens the door to further studies introducing new atmospheric descriptors – capturing the northward shift of Atlantic circulations in winter for instance – or focusing on thermodynamic variables – such as humidity or temperature – in order to fully understand the main mechanisms explaining precipitation trends in the mountainous region of the Northern French Alps. Furthermore, the observed trends in

large-scale circulation over the last 70 years may represent the beginning of stronger and more robust signals, making changes in LSC even more relevant for changes in local weather in the future. The application of the present methodology to GCM projections hence appear as an interesting perspective.

*Code and data availability.* The R code can be requested by email from the corresponding author. The ERA5 reanalysis is available on the Copernicus Climate Data Store (https://cds.climate.copernicus.eu). The ERA20C reanalysis is available at https://apps.ecmwf.int/datasets/

data/era20c-daily/levtype=sfc/type=an/. Informations on how to download the 20CRv2c reanalysis are available at https://psl.noaa.gov/data/gridded/data.20thC_ReanV2c.html. The weather pattern classification have been provided to the authors by Électricité de France for this research. They could be made available to other researchers under a specific research agreement. Requests should be sent to dtg-demande-donnees-hydro@edf.fr.

*Author contributions.* AB: data curation; formal analysis; investigation; methodology; software; visualization; writing-original draft prepa-

490 ration; writing-review and editing. JB: funding acquisition; methodology; project administration; supervision; validation; writing-review and editing. JDC: funding acquisition; methodology; project administration; validation.

*Competing interests.* The authors declare that they have no conflict of interest.

*Acknowledgements.* This study is part of a collaboration between the University Grenoble Alpes and Grenoble Alpes Métropole, the metropolitan authority of the Grenoble conurbation (deliberation 12 of the Metropolitan Council of May 27, 2016). This work is also part

of both the IND-EX project funded by the Rhône Alpes Region, France, and of the HYDRODEMO project financed with the support of the European Union via the FEDER-POIA program and thanks to French state funds via the FNADT-CIMA program.

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
