# Peer review of "Past Evolution of Western Europe Large-scale Circulation and Link to Precipitation Trend in the Northern French Alps"

_Weather and Climate Dynamics, 2021_

## Referee Comment (RC1)

Review of 'Past Evolution and Recent Changes in Western Europe Large-scale Circulation'

In this paper the authors study the non- stationarity of large scale weather patterns (LSCs) over Western Europe, using a set of patterns developed specifically for explaining the differing dynamical origins of French precipitation patterns in Garavaglia (2010). They introduce several metrics which they use to quantify the amplitude, uniqueness and persistence of the large scale circulation patterns in different seasons, and evaluate both the long term representation of these metrics in multiple reanalyses, and more recent trends in ERA5.

Both halves of this work are of interest. The reanalysis intercomparison provides an interesting perspective that clearly highlights some issues with circulation patterns in reanalyses in data-sparse time periods, which perhaps would not be so apparent using more conventional metrics. The analysis of recent trends in circulation presents an interesting approach to exploring non-stationarity in atmospheric dynamics, and shows some interesting shifts in the behaviour of the different LSCs over the last 70 years.

However, in my mind there is a final step which I think would enhance the work quite significantly. As the LSCs used in this paper were developed specifically to explain precip variability, and possible impacts of the LSC changes on precip make up an important part of the conclusions of the paper, it seems a shame that changes in the actual precipitation from 1950-2019 – are not considered. For example, in the conclusion the authors write:

*Mediterranean circulations featuring a marked flow and stationary flow directions that are closely reproduced in the climatology are more frequent over the last 30 years in autumn, which could impact autumn extreme precipitation over the Southwestern Alps.*

I believe showing whether such changes can actually be observed would really strengthen the work, and improve its impact.

Apart from this issue, I believe the manuscript itself needs some slight reworking. Lines 27-54 in the introduction provide a very comprehensive review of the various LSC impacts on Western European weather and extremes. Its actually a very useful collation of these results, but it is incredibly dense and hard to read. Additionally, as there are few links or comparisons made in the paper between the 4 weather types used and these other LSC classifications, a lot of this paragraph is not directly relevant to this work, other than to repeatedly emphasise that LSC has important impacts on surface weather. I would suggest either synthesising these lines down in to a more readable form, or else only including in the main text those studies that are most directly relevant to the work done here, and putting the rest in a helpful reference table (i.e. of weather types against geographical region, with the documented impacts listed in each appropriate cell). I also think better motivation is needed for the various metrics used, due to the fundamentally applied motivation of the work – for example, what does a low singularity mean of an LSC for surface weather? Finally, a bit more work needs to be done in the conclusions to emphasise the implications of the results. For example, at the end of the paper I still don't know what I'm supposed to think about figure 10, or the low significance of the bimodal kdes in figure 9 when compared to the significant unimodal changes of figures 5,7, and 8.

I highlight specific technical issues below, which while quite numerous shouldn't take very long to correct. In summary, I believe this paper represents a valuable piece of work, well suited for inclusion in Weather and Climate Dynamics. I am pleased to see work on circulation pattern non-stationarity, as I consider it to be an understudied but very important area of research. However I am recommending major revisions, so that the authors can include an analysis of the impact of LSC changes on precipitation directly, and to make the more minor revisions to the manuscript I have

suggested. If they would rather not include a direct consideration of precipitation, then I think they need to do a better job of arguing why the changes in these quite regional circulation patterns are interesting enough in their own right.

Specific issues

- I find the use of 'analogy' throughout the paper a bit confusing. It is a matter of personal taste, but I suggest the authors consider using 'analogues' or 'analogue methods' to be clearer.

- Table 2 is not incredibly easy to read and takes up a lot of space. Consider putting it into supplementary material and instead include a table summarising just the significant/interesting differences.

- [line no. 99] ERA20C is in fact a 10 member ensemble. You should verify whether you are using the ens mean or first member, and correct the text.

- [line no. 113] I think more detail is needed here on the weather pattern classification, especially as the approach used is rather atypical. As these patterns are central to this paper, I shouldn't really have to read through all of Garavaglia 2010 to understand what you've done. A few lines explaining that its a hierarchical approach that identifies geopotential patterns associated with rainfall clusters would suffice.

- [line no. 255] If the underlying assimilating model tends to produce calmer, less stormy weather – as most low resolution models do – then the less-constrained reanalysis in the 19$^{th}$ century might be expected to produce calmer weather. This would be consistent with the celerity and singularity trends you find.

- [Line 394] 'Implications for summer heatwaves' - what are the implications? It would be best to state these explicitly.

Technical/editing issues

- [line no. 21] 'Over the large scale' is redundant
- [line no. 41] Something is missing: '..low amplitude through over the UK...'
- line no. 55] 'Over the long run': quite colloquial, better to be more specific – what timescale?
- line no. 55] As this is not a paleo paper is it necessary to refer to the Holocene? At least you should indicate this is the last 10,000 years.
- line no. 73] 'weather pattern'
- line no. 95] identify/determine rather than 'derive'
- line no. 109] 'Studying changes in LSC is carried out': This is not a valid construction. Perhaps 'Changes in LSC are studied using…'?
- [line no. 120] Trends can't really be 'rather poor'. Perhaps say 'small', 'negligible', or 'statistically insignificant'as the case may be
- [line no. 127] A bit more motivation for this score is needed as it is not so common. A brief comment explaining its somewhat similar to using a pattern correlation would help clarify I think.
- [line no. 130] I believe I have worked out this equation now – it is a normalised sum of differences in meridional and zonal gradients at all gridpoints between 2 Z500 maps? Perhaps you could make this a bit clearer. Also 'horizontal and vertical directions' is

misleading, and implies different pressure levels are being considered. 'meridional and zonal directions' would be more precise.
- [Figure 1b] You should make it clear that the black lines are showing trajectories in phase space.
- [line no. 134] 'celerity that is understood as the celerity of deformation...' This is tautological. I suggest 'the speed of deformation' or 'rate of deformation'.
- [line no. 155] 'Even more resembling than usually' is incorrect. I suggest 'Even more similar than usual'. Same for similar errors on lines 308 and 390.
- [line no. 176] 'got' is wrong, I suggest 'obtained'.
- [line no. 240] 'Reduced quantity' rather than 'lower number'?
- [line no. 274] 'found in ERA5' rather than 'thanks to ERA5'?
- [Figure 5] 'sup-periods'

---

## Referee Comment (RC2)

**Review of "Past evolution and recent changes in Western Europe large-scale circulation" by Blanc et al. submitted to Weather and Climate Dynamics (WCD-2021-69)**

**General comments**

This study investigates the evolution of geopotential height over Central Europe during the past 170 years based on different reanalyses. The authors first analyze how similar the geopotential height fields are in terms of shape among the different reanalysis products. They further investigate how intrinsic characteristics of large-scale atmospheric states such as stationarity, uniqueness, and gradients compare between different reanalyses and how they developed over the past 170 years. They ultimately stratify these flow characteristics according to the four major Central European flow patterns. The findings are used to provide potential explanations for changes in Central European surface weather over the past century found in previous studies. The manuscript has a clear structure, the underlying method is simple and relatively elegant, and the various methodological steps have been performed and described thoroughly. Overall, the findings of this study are important to further our understanding of historic (and thus potentially future) changes in Central European weather. Furthermore, they are valuable for users of any of these reanalysis products because they highlight the partly substantial differences between these "observational" datasets that are often treated as the "truth". Despite these valuable insights, the study is quite detailed, descriptive, and thus on the edge of being too lengthy. Furthermore, it partly misses to better highlight the novelty of its findings and how they can specifically be used for further research. Also, some of the methodological steps and figures require more careful description, particularly for readers who are not familiar with the used method. Therefore, I recommend publishing this manuscript after major revisions.

**Major comments**

Interpretability and implications of the results (1): Although the method (TWS and derived descriptors) is elegant in my view, I struggle to interpret some of the findings obtained with it and the implications it has for further research. For instance, Fig. 4 is very interesting and contains lots of important information and food for thoughts. Nevertheless, it is hard (at least to me) to "translate" the differences between reanalyses and between periods into more intuitive concepts of the large-scale circulation (i.e., geopotential height, storm track, weather regimes, etc.). For instance, what does a change in the summer celerity anomaly from –0.06 to 0.0 (in 20CR mean) over the last 170 years mean? Is this a large change in stationarity we would for instance see in weather regime statistics as well? And should we worry about these differences when using these reanalysis products for certain applications (and for which applications)? Likewise, what does the large difference in singularity and relative singularity in the early 20[th] century between 20CR and ERA20C tell us? Which of the two reanalyses should we "trust" more? Can you discuss this in your manuscript, if possible? Related to this, I think you should briefly discuss the limitations of the TWS in general. For instance, it does not tell us how exactly two geopotential fields differ from each other. You partly overcome this by additionally showing differences in geopotential fields (e.g., Fig. 3), but you could still discuss more clearly what added value we get from using this score and what limitations it has.

Interpretability and implications of the results (2): I assume the multi-annual/-decadal fluctuations in the different atmospheric descriptors (Figs. 2 and 4) is caused by both trends/changes in the observations that are assimilated and multi-annual/-decadal variability in the Atlantic-European large-scale state (for instance, a decade of predominantly persistent negative NAO phases might manifest in a lower celerity; cf., e.g., Woollings et al., 2015). Can you estimate or at least speculate which of the two might be more

important, particularly in the late 19$^{th}$ and early 20$^{th}$ centuries? Have other studies investigated this question? It might be a quite relevant question for users of the different reanalysis products.

Significance testing: You perform careful and important significance tests for detecting trends/changes in the atmospheric descriptors, which is great, but you don't do the same for the differences in the geopotential anomalies shown in, e.g., Fig. 6. Could you perform such a test and highlight the significant regions? For instance, I doubt that the mentioned reinforcement of the Atlantic anticyclone in winter (L303) in Fig. 6 (top left) is really significant…

Investigated domain: I understand the reason for using the study region centered over Central and Southern Europe (rectangle in Fig. 1a), as you are primarily interested in focusing on weather over France. However, I think some of your results for the descriptors might be quite sensitive to the choice of this domain. Did you test this? For instance, I assume increasing the region over whole Europe or even including the North Atlantic might yield results that are more linked to the typical weather regimes in these regions and might thus be easier to interpret. If you do not want to discuss this in too much detail in the manuscript, could you at least extend the anomaly maps in Figs. 1a, 3, and particularly 6 to a larger domain including the main centers of action of the NAO and the further modes of variability (of course by leaving the black rectangles to see the study region)?

Method: It took me some time to fully understand the TWS and the derived descriptors (I did not know it before), which I assume other readers might experience, too. Although your schematic in Fig. 1b is very helpful, there are some methodological details you could add to the text. For instance, on L176: How exactly do you compute the TWS between two reanalyses when referring to Eq. 1? What are the days $t_k$ and $t_{k'}$ in this context? Do you go through each day (for instance in a specific season) and compute the corresponding TWS between the field of reanalysis 1 and reanalysis 2, which gives you one TWS value for each day, which you then average over the season? This might be a trivial question to you, but it is not obvious from considering Eq. 1, which refers to a day k and another day k'. Furthermore, relating to Fig. 4: Again, I am not sure if I completely understand the details here. Let me explain how I understand it: for the celerity of a specific season between 1850 and 2010 (e.g., for 20CR), you go through each day k during each DJF season and compute the celerity with respect to the previous day k-1. This yields as many celerity values as you have days in this time period. You finally compute a 5-year running mean over this daily celerity time series, which is plotted in Fig. 4 (as anomalies relative to the celerity climatology between 2006 and 2010). Is this correct? Likewise, for the singularity and relative singularity you go through each day and compute the two terms by using a Q of 0.5% of all days available? Are the 0.5% relative to the whole period 1850 to 2010, or do you split it in sub-periods as you mention on L160? And finally, why do you compute the index anomalies against a climatology between 2006 and 2010 only? Would it not be much more robust to use the climatology over the whole investigated period?

**Minor comments**

Title: I would write "in Western European large-scale circulation", but I'm not fully sure.

L20: You could slightly extend this sentence in the sense that the LSC provides the general dynamical setting for various local weather phenomena, not just via the airflow towards a region but also via modifying stability and moisture availability.

L21: Replace "Over the large scale" with "The"

L23: This sounds a bit odd, because the further (second, third...) modes of variability are also active throughout the year, i.e., not just the NAO.

L25: "by intensifying westerlies"

L25-27: This is an important point, that additional modes of variability are needed to explain surface weather modulation particularly in Central Europe. You could add some further references here, for instance Pasquier et al. (2019), who investigated the link between precipitation extremes in Europe and a higher number of weather regimes than the classic 4 regimes.

L28: "associated with wet conditions at the northern flank"

L38: Rather something like "low pressure systems developing over the Atlantic and Iberian Peninsula"?

L39: "associated with strong southwesterly to southerly flows"?

L43-45: I would distinguish here between stationary cyclones that lead to persistent moisture transport towards the Alps and thus accumulated precipitation and large-scale situations associated with high instability (for instance induced by upper-level cut-off lows; e.g., Portmann et al., 2021) and thus strong forcing for thunderstorms, which are also crucial for (convective) heavy precipitation.

L49: Remove "long-lasting" as it is already stated with "persistence".

L51-52: You could for instance add Zschenderlein et al. (2019) who investigated the processes behind European heat waves in detail and climatologically.

L55: "may have been linked to"

L88: "500 hPa geopotential height ranges from"

L89: Rather "pressure distribution"?

L93: How exactly is the ensemble for the 20CRv2c created, i.e. which kinds of "perturbations" are they based on?

L96: Could you add one or two sentences about the quality of this reanalysis, for instance during the 19[th] century? I assume continuous pressure / SST / sea ice observations from ships were rather rare at that time?

L103: "referred to as full-input"

L112 and later: I'm not so convinced about the term "atmospheric influences". As they basically refer to the direction of the flow towards Southern France, could you replace it with something like "flow directions", "flow patterns", "weather patterns", or similar? Also, you might switch 3.1 and 3.2, because you mainly start by analyzing the atmospheric descriptors and only then use the atmospheric influences. If done so, you could also split Fig. 1 into two, with a first figure showing the schematic in current Fig. 1b, and a second figure showing the patterns in current Fig. 1a. Furthermore, I would find it very useful to see the links of these four patterns to well-known North Atlantic-European weather regimes / modes of variability. You could partly achieve this by increasing the plotted domain (see one of my major comments). Furthermore, you could write a few sentences about it or refer to other studies that did.

Fig. 1a: Could you increase the wind vectors in Fig. 1a, or make them thinner? It is hard to see their directions. Furthermore, are the names Northeast and Anticyclonic really appropriate, as you primarily consider the influence of the pattern on the flow towards Southern France? Would it not rather be something like Northern (for Northeast) and Northeast (for Anticyclonic)?

L120: "trends being poor" sounds odd – you should rephrase to "small".

L178: "to allow for the computation"

L162: Why do these three measures only focus on shape? As far as I understand the TWS, it considers the (sum of the) pressure difference of each grid point with respect to each other grid point in a certain domain, and thus indirectly also accounts for differences in pressure magnitudes, right?

L166: I am a bit confused about the notation in Eq. 5: Does max_j stand for the maximum value of all grid points j within the rectangle domain? So MPD is the difference of the maximum pressure and minimum pressure (one grid point each) on a certain day k?

L168: I assume the weak relation of your MPD to the NAO is simply because your rectangular domain (Fig. 1a) is too small to include the centers of action of the NAO, right? So, if you would increase your domain enough a large MPD would start to coincide with a strong positive or negative NAO, right (see also one of my major comments)?

L170: What exactly do you mean with "per season"? Do you compute all four atmospheric predictors (cel, sing, rsing, mpd) on a daily basis and you then average them over the seasons, or do you compute the atmospheric predictors for the already averaged seasonal geopotential fields? I guess it must be the former, but you should specify this.

L183: At first sight, it seems counter-intuitive that the geopotential fields agree less well in summer than in winter, considering the generally larger baroclinic activity in the investigated region in winter probably associated with a larger potential for "errors" in the reanalysis. Do you have an idea why they agree less well in summer? Could it be associated with (smaller-scale) convection?

L188: Why are the shapes more similar between 1850 and 1900 than between 1900 and 1950? I would have assumed that the shapes become generally less similar the fewer observations are available (i.e., the further back you go in time), but this is not the case. Is it because you generally increase the degrees of freedom from 1900 on, with the introduction of further types of observations and the 4D-Var? Can you explain?

L194: Did you plot the same as in Fig. 3 but with the absolute fields (i.e., 4 fields for each season: 20CR late-century, 20CR early-century, ERA20C late-century, ERA20C early-century)? If yes, could you show them in your response?

L210: Does this mean that 20CR should only be used with caution for studying any climate change / trends over the past 170 years, as it misses a dynamically very relevant change of meridional pressure gradients over the Atlantic-European region (assuming that ERA5 is the best product we have, which is a reasonable assumption)?

L217-219: I find it intuitive that the higher resolution and "more details" in ERA5 generally increase the MPD compared to the other reanalyses, but I do not find it obvious for the other descriptors. Wouldn't a higher celerity for instance mean generally less persistent atmospheric states? How do you explain this? Is it because you get more smaller-scale features in ERA5 which tend to be less persistent? Likewise, does the much higher relative singularity in ERA5 imply much more rare/extreme atmospheric states in ERA5 compared to the other reanalyses? Why is this – due to the same reason as above? I think it would be helpful for the reader if you could discuss this a bit.

L274: "according to ERA5" or something similar

Figs. 5, 7, and 8: These are nice figures, but they contain a lot of information. To simplify slightly, would it make sense to replace the classic box plots (right and left of the difference) with violin plots but remove the difference plots instead? You could even make the blue and red violins transparent and overlay them, which would allow directly seeing the absolute values and differences. This would reduce the figure from currently 48 boxes to 16. Could you test this and check whether it is useful?

Fig. 5: I'm not sure if I understand correctly: You say that the percentiles are defined with respect to all days within 1950 and 2019. So, shouldn't the sum of the blue and red bars consequently give a distribution that is somewhat centered around the 50$^{th}$ percentile on the y-axis? For many descriptors, such as the celerity in winter or summer, however, this does not seem to be the case. Why is that?

L296: "meaning new anticyclonic patterns are less explored over the present period" – this sounds a bit strange, although I know what you mean referring to the singularity. It sounds like we had completely

different atmospheric patterns in the earlier period, which do not exist anymore. Can you rephrase somehow?

L300: "associated with high pressure"

L310: "reproducibility of Atlantic circulations" sounds a bit like a model that fails to reproduce these patterns, although we are dealing with reanalysis. Could you rephrase somehow?

Figs. 9 and 10: These two figures appear out of a sudden in Section 4.2.3, without introducing the reasons behind showing them (for instance, why only for summer and autumn, and the Mediterranean circulation?). Furthermore, the description in the caption of Fig. 10 is hard to understand and I had to read it several times. In general, I wonder if these two figures are really needed or whether you could simply state some of the most important changes in, for instance, the occurrence frequency of the most extremely stationary states? Although the idea behind Fig. 9 is good, for instance, I find it hard to grasp a message from looking at this figure, as the two distributions are overall rather similar. Removing these two figures would further help to reduce the content of the rather long manuscript…

L337-339: What might be the physical reason to obtain such a "seasonal shift" as you call it (considering the fact that the solar cycle, i.e. the main driver of the seasonal cycle, should not be affected by any change)? Isn't it more the northward shift of the storm track that we might see here, which influences the Mediterranean differently during the different seasons because its overall year-round meridional oscillation also changes with the northward shift?

Conclusions: Coming back to one of my major comments, I miss some more specific implications / ideas for further research in this last paragraph. When reading the conclusions, it is not clear whether your main take-home message should be the usefulness of your method to better understand surface weather extremes in France, or whether it is more about better understanding historic large-scale changes over Europe in general with this special approach and highlighting differences in reanalysis products. To me, it is kind of both, but you should state/distinguish this more clearly. Also, you could again refer some of your results to some other specific studies or ongoing debates. For instance, you could relate your findings about celerity (i.e., stationarity) to the widely discussed (but partly disputed) increase in jet stream waviness due to Arctic amplification, potentially leading to more persistent and thus extreme weather (e.g., Kornhuber & Tamarin-Brodsky, 2021).

**References**

Kornhuber & Tamarin-Brodsky, 2021: Future changes in Northern Hemisphere summer weather persistence linked to projected Arctic warming. *Geophysical Research Letters*, https://doi.org/10.1029/2020GL091603

Pasquier et al., 2019: Modulation of atmospheric river occurrence and associated precipitation extremes in the North Atlantic Region by European weather regimes. *Geophysical Research Letters*, https://doi.org/10.1029/2018GL081194

Portmann et al., 2021: The three-dimensional life cycles of potential vorticity cutoffs: a global and selected regional climatologies in ERA-Interim (1979–2018), *Weather Clim. Dynam.*, https://doi.org/10.5194/wcd-2-507-2021

Woollings et al., 2015: Contrasting interannual and multidecadal NAO variability. *Clim Dyn,* https://doi.org/10.1007/s00382-014-2237-y

Zschenderlein et al., 2019: Processes determining heat waves across different European climates. *Q J R Meteorol Soc,* https://doi.org/10.1002/qj.3599

---

## Author Response (AR1)

**Referee #1**

We sincerely thank Referee #1 for very valuable feedbacks.

Here are the main modifications that will be considered following Referee's comments. We will consider the study of precipitation in the Northern French Alps in the paper, as the atmospheric descriptors were designed for this specific purpose. The consideration of seasonal and extreme precipitation trends in the Northern French Alps in section 4.3 will allow for more concrete and clear interpretations on the implications of large-scale circulation changes for local weather. This will also support the use of the atmospheric descriptor over the Western Europe domain to study large-scale circulation trends. In this way, the introduction and conclusions will be more specific on Alpine precipitation. The Atlantic and Mediterranean influences will be the only atmospheric influences considered in the paper as Anticyclonic conditions and Northeast circulations are not relevant for seasonal and extreme precipitation in the Northern French Alps (Fig.5 will be removed). Fig. 9 and Fig.10 will be replaced by figures providing more relevant information on LSC driving precipitation in the Northern French Alps. Furthermore, we will add in Fig.3 the maps of the differences in 500 hPa geopotential height between 20CRv2c and ERA20C for both the period 1900-1930 and 1970-2000. This will allow for a more physical interpretation of the differences in geopotential shapes exposed in Fig.2.

Please find below a detailed point-to-point reply to the Referee's comments.

**General comments**

*In this paper the authors study the non- stationarity of large scale weather patterns (LSCs) over Western Europe, using a set of patterns developed specifically for explaining the differing dynamical origins of French precipitation patterns in Garavaglia (2010). They introduce several metrics which they use to quantify the amplitude, uniqueness and persistence of the large scale circulation patterns in different seasons, and evaluate both the long term representation of these metrics in multiple reanalyses, and more recent trends in ERA5. Both halves of this work are of interest. The reanalysis intercomparison provides an interesting perspective that clearly highlights some issues with circulation patterns in reanalyses in data-sparse time periods, which perhaps would not be so apparent using more conventional metrics. The analysis of recent trends in circulation presents an interesting approach to exploring non-stationarity in atmospheric dynamics, and shows some interesting shifts in the behaviour of the different LSCs over the last 70 years. However, in my mind there is a final step which I think would enhance the work quite significantly. As the LSCs used in this paper were developed specifically to explain precip variability, and possible impacts of the LSC changes on precip make up an important part of the conclusions of the paper, it seems a shame that changes in the actual precipitation from 1950-2019 – are not considered. For example, in the conclusion the authors write: Mediterranean circulations featuring a marked flow and stationary flow directions that are closely reproduced in the climatology are more frequent over the last 30 years in autumn, which could impact autumn extreme precipitation over the Southwestern Alps. I believe showing whether such changes can actually be observed would really strengthen the work, and improve its impact.*

⇒ Thanks for this very interesting feedback. Whether or not to consider precipitation in this paper has been actually widely discussed with the co-authors. The atmospheric descriptors were indeed developed to study the large-scale circulations of interest for extreme precipitation in the Northern French Alps (Blanchet et al., 2018; Blanchet and Creutin, 2020; Blanc et al., 2021a). In

the first draft of this paper, we decided to consider large-scale circulations (LSC) only to make it more focused, as studying the impact of LSC changes on precipitation represents substantial additional material. However, we understand and agree with your point, so in this second draft we will include the study of precipitation. We will add a third subsection in the results in which we will study trends in seasonal and extreme precipitation in two medium size catchments of the Northern French Alps (Isère and Drac River catchment at Grenoble), and where we will discuss the links with LSC changes. We hope that the current version of the paper will represent a more complete piece of work.

*Apart from this issue, I believe the manuscript itself needs some slight reworking. Lines 27-54 in the introduction provide a very comprehensive review of the various LSC impacts on Western European weather and extremes. Its actually a very useful collation of these results, but it is incredibly dense and hard to read. Additionally, as there are few links or comparisons made in the paper between the 4 weather types used and these other LSC classifications, a lot of this paragraph is not directly relevant to this work, other than to repeatedly emphasise that LSC has important impacts on surface weather. I would suggest either synthesising these lines down in to a more readable form, or else only including in the main text those studies that are most directly relevant to the work done here, and putting the rest in a helpful reference table (i.e. of weather types against geographical region, with the documented impacts listed in each appropriate cell). I also think better motivation is needed for the various metrics used, due to the fundamentally applied motivation of the work – for example, what does a low singularity mean of an LSC for surface weather? Finally, a bit more work needs to be done in the conclusions to emphasise the implications of the results. For example, at the end of the paper I still don't know what I'm supposed to think about figure 10, or the low significance of the bimodal kdes in figure 9 when compared to the significant unimodal changes of figures 5,7, and 8.*

⇒ Thanks for this valuable comment. As the new version of the paper will consider precipitation, the whole article will be redirected with a more specific focus on precipitation in the Alpine range. The introduction will be refined, considering previous studies that linked large-scale circulation to precipitation variability and extremes in the Alpine range or in France. References linking large-scale circulation to local temperature or to different variables in other regions than the Alps will be removed. We will include a full paragraph detailing the links between the atmospheric descriptors (celerity, singularity, relative singularity and Maximum Pressure Difference) and precipitation variability and extremes in the Northern French Alps – so many reasons that motivate the study of LSC changes using these descriptors. In the same vein, Figures 9 and 10 will be replaced by figures focusing more specifically on descriptors values that are relevant for precipitation variability and extremes. In this way, the implications of LSC changes for precipitation in the Northern French Alps will be clearer and the associated conclusions will be more clearly exposed.

*I highlight specific technical issues below, which while quite numerous shouldn't take very long to correct. In summary, I believe this paper represents a valuable piece of work, well suited for inclusion in Weather and Climate Dynamics. I am pleased to see work on circulation pattern nonstationarity, as I consider it to be an understudied but very important area of research. However I am recommending major revisions, so that the authors can include an analysis of the impact of LSC changes on precipitation directly, and to make the more minor revisions to the manuscript I*

*have suggested. If they would rather not include a direct consideration of precipitation, then I think they need to do a better job of arguing why the changes in these quite regional circulation patterns are interesting enough in their own right.*

⇒ Thanks for pointing the relevance of this area of research. As said above, the new manuscript will consider precipitation. The specific technical issues will also be addressed.

**Specific issues**

- *I find the use of 'analogy' throughout the paper a bit confusing. It is a matter of personal taste, but I suggest the authors consider using 'analogues' or 'analogue methods' to be clearer.*

  ⇒ Thanks for this feedback. The term "analogy" is explained in more details when introducing the atmospheric descriptors, which is probably too late for a clear understanding. We have therefore decided to include the following explanation directly in the abstract: "We focus on the evolution of large-scale circulation characteristics using three atmospheric descriptors that are based on analogy, by comparing daily geopotential height fields to each other". From our point of view, the use of "analogue method" refers more to considering analog days for reconstructing local variables (such as daily precipitation or temperature) on a given region based on large-scale predictors. This is not the case here, as the analogy is only used to construct atmospheric descriptors that characterize large-scale circulation. Thus we propose to keep the term "analogy" but we hope that the explanation of the concept already in the abstract will make it easier to understand.

- *Table 2 is not incredibly easy to read and takes up a lot of space. Consider putting it into supplementary material and instead include a table summarising just the significant/interesting differences.*

  ⇒ Thanks for this comment. In this new version, we will focus on LSC that are relevant for precipitation in the Northern French Alps. Thus we will only consider the Atlantic and the Mediterranean influences and we will no longer consider the Anticyclonic and Northeast influences. Atlantic and Mediterranean influences have been shown to drive both seasonal and extreme precipitation in the Northern French Alps (Blanc et al., 2021a; Blanchet et al., 2021; Sodemann and Zubler, 2010). Table 2 will be therefore much shorter, and it will be transposed to take much less space.

- *[line no. 99] ERA20C is in fact a 10 member ensemble. You should verify whether you are using the ens mean or first member, and correct the text.*

  ⇒ A 10-member ensemble is indeed used in the construction of ERA20C. It is based on different evolution of SST and sea-ice and it considers model errors as well as uncertainties in the assimilated observations. However, this ensemble is only used to derive spatial and temporal errors that are then used as an input in the main reanalysis product, which is single-member. The most recent version of ERA20C we use (the one of Poli et al., 2016) follows the methodology of ERA20 deterministic (2015), which is itself based on the 10-member reanalysis (2013) (https://www.ecmwf.int/en/elibrary/11700-era-20c-deterministic). The construction of the most recent version of ERA20C is detailed in Poli et al. (2016), whose abstract clearly states that "The reanalysis is single-member, and the background errors are spatiotemporally varying, derived from an ensemble".

- *[line no. 113] I think more detail is needed here on the weather pattern classification, especially as the approach used is rather atypical. As these patterns are central to this paper, I shouldn't really have to read through all of Garavaglia 2010 to understand what you've done. A few lines explaining that its a hierarchical approach that identifies geopotential patterns associated with rainfall clusters would suffice.*

  ⇒ Thanks for this comment. That is true; we will add some explanations about the construction of the classification.

- *[line no. 255] If the underlying assimilating model tends to produce calmer, less stormy weather – as most low resolution models do – then the less-constrained reanalysis in the 19th century might be expected to produce calmer weather. This would be consistent with the celerity and singularity trends you find.*

  ⇒ Thanks for this comment. That is true, the different results from 20CR suggest that the observed trends in celerity and singularity in the second half of the 19th century are more an artefact of the data set rather than physical signals. This is what we intended to say in this sentence, in the sense that we could not physically interpret such trends. But we agree the sentence was actually not clear enough, so it will be rewritten.

- *[Line 394] 'Implications for summer heatwaves' - what are the implications? It would be best to state these explicitly.*

  ⇒ Thanks for this comment. We have decided to exclude trends in Anticyclonic conditions in the new version of the paper, as i) Anticyclonic conditions are not associated with precipitation in the two medium size mountainous catchments we study (Blanchet et al., 2021; Blanc et al., 2021b), ii) this influence features only slight changes compared to the Mediterranean and the Atlantic influences, and iii) considering trends in Anticyclonic conditions in addition to precipitation would lead to a long article. Thus, the corresponding sentence will be removed.

**Technical/editing issues**

- *[line no. 21] 'Over the large scale' is redundant*

  ⇒ Will be corrected.

- *[line no. 41] Something is missing: '..low amplitude through over the UK...'*

  ⇒ Will be corrected. That was actually a typing error, as we were referring to a trough over the UK.

- *line no. 55] 'Over the long run': quite colloquial, better to be more specific – what timescale?*

  ⇒ This part of the introduction will be removed as it was not essential for setting the context and the problematic.

- *line no. 55] As this is not a paleo paper is it necessary to refer to the Holocene? At least you should indicate this is the last 10,000 years.*

  ⇒ Same answer as the previous comment.

- *line no. 73] 'weather pattern'*

  ⇒ Will be corrected.

- *line no. 95] identify/determine rather than 'derive'*

  ⇒ Will be corrected.

- *line no. 109] 'Studying changes in LSC is carried out': This is not a valid construction. Perhaps 'Changes in LSC are studied using. . . '?*

  ⇒ Will be corrected, thanks.

- *[line no. 120] Trends can't really be 'rather poor'. Perhaps say 'small', 'negligible', or 'statistically insignificant'as the case may be*

  ⇒ Will be corrected, thanks for this comment.

- *[line no. 127] A bit more motivation for this score is needed as it is not so common. A brief comment explaining its somewhat similar to using a pattern correlation would help clarify I think.*

  ⇒ Thanks. The TWS score has the advantage of considering only the similarity in shape of geopotential height fields, whatever the absolute height of the geopotential. The shape of the geopotential defines the flow direction which is relevant for precipitation, especially in mountainous regions (Blanchet et al., 2021; Horton et al., 2012). This score is widely employed in the analog method to reconstruct precipitation based on analogy in geopotential height fields (Daoud et al., 2016; Marty et al., 2012; Wetterhall et al., 2005). We will highlight the relevance of the score for precipitation and we will cite these references.

- *[line no. 130] I believe I have worked out this equation now – it is a normalised sum of differences in meridional and zonal gradients at all gridpoints between 2 Z500 maps? Perhaps you could make this a bit clearer. Also 'horizontal and vertical directions' is misleading, and implies different pressure levels are being considered. 'meridional and zonal directions' would be more precise.*

  ⇒ Thanks for this comment. You're right, that's exactly what the TWS score represents. We will take your suggestions into account to clarify the description of the score.

- *[Figure 1b] You should make it clear that the black lines are showing trajectories in phase space.*

  ⇒ Will be corrected, thanks.

- *[line no. 134] 'celerity that is understood as the celerity of deformation...' This is tautological. I suggest 'the speed of deformation' or 'rate of deformation'.*

  ⇒ Will be corrected, thanks for this comment.

- *[line no. 155] 'Even more resembling than usually' is incorrect. I suggest 'Even more similar than usual'. Same for similar errors on lines 308 and 390.*

  ⇒ Will be corrected, thanks for this comment.

- *[line no. 176] 'got' is wrong, I suggest 'obtained'.*

  ⇒ Will be corrected, thanks.

- *[line no. 240] 'Reduced quantity' rather than 'lower number'?*

  ⇒ Will be corrected, thanks.

- *[line no. 274] 'found in ERA5' rather than 'thanks to ERA5'?*

  ⇒ We used "thanks to ERA5" because this reanalysis allows us to extend the analysis until 2019. But this is not essential to the sentence, so we will use "according to ERA5".

- *[Figure 5] 'sup-periods'*

  ⇒ Will be corrected, thanks.

**Referee #2**

We sincerely thank Referee #2 for very valuable feedbacks.

Here are the main modifications that will be considered following Referee's comments. We will consider the study of precipitation in the Northern French Alps in the paper, as the atmospheric descriptors were designed for this specific purpose. The consideration of seasonal and extreme precipitation trends in the Northern French Alps in section 4.3 will allow for more concrete and clear interpretations on the implications of large-scale circulation changes for local weather. This will also support the use of the atmospheric descriptor over the Western Europe domain to study large-scale circulation trends. In this way, the introduction and conclusions will be more specific on Alpine precipitation. The Atlantic and Mediterranean influences will be the only atmospheric influences considered in the paper as Anticyclonic conditions and Northeast circulations are not relevant for seasonal and extreme precipitation in the Northern French Alps (Fig.5 will be removed). Fig. 9 and Fig.10 will be replaced by figures providing more relevant information on LSC driving precipitation in the Northern French Alps. Furthermore, we will add in Fig.3 the maps of the differences in 500 hPa geopotential height between 20CRv2c and ERA20C for both the period 1900-1930 and 1970-2000. This will allow for a more physical interpretation of the differences in geopotential shapes exposed in Fig.2.

Please find below a detailed point-to-point reply to the Referee's comments.

**General comments**

*This study investigates the evolution of geopotential height over Central Europe during the past 170 years based on different reanalyses. The authors first analyze how similar the geopotential height fields are in terms of shape among the different reanalysis products. They further investigate how intrinsic characteristics of large-scale atmospheric states such as stationarity, uniqueness, and gradients compare between different reanalyses and how they developed over the past 170 years. They ultimately stratify these flow characteristics according to the four major Central European flow patterns. The findings are used to provide potential explanations for changes in Central European surface weather over the past century found in previous studies. The manuscript has a clear structure, the underlying method is simple and relatively elegant, and the various methodological steps have been performed and described thoroughly. Overall, the findings of this study are important to further our understanding of historic (and thus potentially future) changes in Central European weather. Furthermore, they are valuable for users of any of these reanalysis products because they highlight the partly substantial differences between these "observational" datasets that are often treated as the "truth". Despite these valuable insights, the study is quite detailed, descriptive, and thus on the edge of being too lengthy. Furthermore, it partly misses to better highlight the novelty of its findings and how they can specifically be used for further research. Also, some of the methodological steps and figures require more careful description, particularly for readers who are not familiar with the used method. Therefore, I recommend publishing this manuscript after major revisions.*

**Major comments**

*Interpretability and implications of the results (1): Although the method (TWS and derived descriptors) is elegant in my view, I struggle to interpret some of the findings obtained with it and*

*the implications it has for further research. For instance, Fig. 4 is very interesting and contains lots of important information and food for thoughts. Nevertheless, it is hard (at least to me) to "translate" the differences between reanalyses and between periods into more intuitive concepts of the large-scale circulation (i.e., geopotential height, storm track, weather regimes, etc.). For instance, what does a change in the summer celerity anomaly from –0.06 to 0.0 (in 20CR mean) over the last 170 years mean? Is this a large change in stationarity we would for instance see in weather regime statistics as well? And should we worry about these differences when using these reanalysis products for certain applications (and for which applications)? Likewise, what does the large difference in singularity and relative singularity in the early 20th century between 20CR and ERA20C tell us? Which of the two reanalyses should we "trust" more? Can you discuss this in your manuscript, if possible? Related to this, I think you should briefly discuss the limitations of the TWS in general. For instance, it does not tell us how exactly two geopotential fields differ from each other. You partly overcome this by additionally showing differences in geopotential fields (e.g., Fig. 3), but you could still discuss more clearly what added value we get from using this score and what limitations it has.*

⇒ Thanks for this comment.

**Interpretability:** The descriptors used in this study allow characterizing in a quantitative manner every daily large-scale circulation (LSC). Thus they allow studying trends such as in Fig. 4, which offers a different view on LSC trends in comparison to weather pattern occurrence or storm track activity.

The descriptors have been designed to study LSC driving extreme precipitation in the Northern French Alps (Blanc et al., 2021a; Blanchet et al., 2018; Blanchet and Creutin, 2020). The descriptor study showed that LSC driving extreme precipitation in the Northern French Alps share some common characteristics, whatever the atmospheric influence.

We agree that taken alone, the singularity and relative singularity can be difficult to translate into a more physical representation. However, some links between these descriptors and circulation patterns have been exposed in Blanc et al. (2021a). Notably, the singularity of a given Western Europe LSC appears to be related to the zonality of the flow – the more zonal LSC being the least singular. We will add a comment on that in the new version of the paper when interpreting Fig. 4.

Furthermore, we think that the following points – based on both the submitted version of the paper and future modifications – will answer most of your comment about results interpretation:

- According to our results, the main conclusion of section 4.1 is that LSC differ in the different reanalyses products before 1950, and that trends before this date should be taken with caution. A more physical interpretation of the descriptor trends is provided in section 4.2 over the period 1950-2019, over which the different reanalyses agree;

- The magnitude of the differences between reanalyses from 1851 to 2010 is assessed in Fig. 2. In order to allow for a more physical interpretation of a difference in TWS, this figure already shows an horizontal line at 0.28 which corresponds to the TWS between day D and day D-1 mapped in Fig. 1. We will add a second horizontal line ($TWS = 0.18$) which corresponds to the TWS between day D and another day mapped in Fig.1. This will allow for a better physical representation of the differences between reanalyses. The TWS scores of 0.28 and 0.18 will also be explicitly written in Fig.1 next to the corresponding arrow;

- Fig. 2 and Fig. 4 are complemented with the maps of Fig. 3, which illustrates geopotential

height differences between two sub-periods. These maps allow for a more physical interpretation of the results obtained using the atmospheric descriptors. However, this is true that the 500 hPa geopotential height difference between 20CRv2c and ERA20C is not represented. We will add such maps to Fig.3, both for the period 1900-1930 and 1970-2000;

- In Figure 4, the trend differences in descriptor values before 1950 are out of the range of the descriptor natural variability (as commented l.224). This large *signal to noise* ratio gives an information on the significant differences between reanalyses.

Furthermore, we cannot say which reanalysis is to be trusted more. Such a conclusion would require comparing reanalyses with independent observations, which is out of the scope of the present paper. However, we discuss several times in section 4.1 the potential causes of such differences between reanalyses based on the recent literature.

**Implications of the results:** Reanalyses are often considered as truth, as they are based on observations. Here, we show that different reanalyses lead to different trends in LSC along the 20th century. So the main implication of the results shown in section 4.1 is that trends in Western Europe LSC before 1950 must be taken with caution. Following your comment, we will highlight this more.

**TWS:** The Teweles-Wobus score (Teweles and Wobus, 1954) measures the similarity in shape between geopotential height fields. The geopotential shape defines the flow direction which is relevant for regional precipitation, especially in mountainous regions (Blanchet et al., 2021; Horton et al., 2012). TWS is therefore widely used in the analog method to reconstruct precipitation based on geopotential height fields (Daoud et al., 2016; Marty et al., 2012; Wetterhall et al., 2005). The TWS between day $k$ and day $k'$ represents a normalized sum of differences in meridional and zonal gradients at all gridpoints over Western Europe between the 500 hPa geopotential height field of day $k$ and day $k'$. Therefore it measures a normalized difference in geopotential shape, i.e. a difference in shape that is independent of both the absolute geopotential height and the absolute pressure gradient over the region of study. One limitation of TWS is that is can produce large values (i.e. different shapes) for quite flat geopotential shapes, which therefore represent small absolute differences in LSC. Such an example is given in Fig. 1. The summer and autumn seasons are more concerned, as they feature the flattest geopotential shapes (see the 0.1 quantile of MPD in Fig 2, Blanc et al., 2021a). This is why we also consider the Maximum Pressure Difference (MPD) which accounts for the absolute pressure gradient.

We will add more information about the construction of TWS in the paper. We will also discuss in Section 4.1 the limitations of the score especially in summer when interpreting the differences in geopotential shapes.

*Interpretability and implications of the results (2): I assume the multi-annual/-decadal fluctuations in the different atmospheric descriptors (Figs. 2 and 4) is caused by both trends/changes in the observations that are assimilated and multi-annual/-decadal variability in the Atlantic-European large-scale state (for instance, a decade of predominantly persistent negative NAO phases might manifest in a lower celerity; cf., e.g., Woollings et al., 2015). Can you estimate or at least speculate which of the two might be more important, particularly in the late 19th and early 20th centuries? Have other studies investigated this question? It might be a quite relevant question for users of the*

[Figure]

Figure 1: 500 hPa geopotential height of day 02/07/1961 and day 03/07/1961 over Europe, according to ERA5. The arrows represent the horizontal wind at 500 hPa. The black rectangle represents the domain considered to compute the TWS. The TWS between the two days – which we define as the celerity of day 03/07/1961 – reaches 0.34. This corresponds to the 92 % percentile of celerity, that is to a large TWS between consecutive days. The MPD of day 03/07/1961 is of 117 m which corresponds to the 0.3 % percentile of MPD, that is a very flat geopotential shape. The TWS score, which is a normalized score, produces a large value because of the differences in geopotential shapes in the center and upper half of the domain. This can only be seen with the different arrow directions, but not when considering geopotential height alone with such color classes. In this case, the geopotential shape in the sense of TWS is quite different while the absolute flow is not. This makes the TWS less relevant for very flat geopotential shapes.

[Figure]

Figure 2: Extracted from Fig. 7 of Blanc et al. (2021a). Interannual median (blue), 0.1 and 0.9 quantiles (grey) of MPD for the period 1950–2017.

*different reanalysis products.*

⇒ Thanks for this comment. The similar fluctuations of descriptor values between reanalyses suggest that these fluctuations come from natural variability. It is quite difficult to relate these fluctuations to larger fluctuations in the whole Europe/North-Atlantic domain because we focus on a much smaller region over Western Europe, and the relation between our descriptors and NAO appears to be weak (Blanc et al., 2021b). Our answer to your fourth major comment will provide insights about the relevance of the considered domain over Western Europe for our study.

Furthermore, the differences in trends between the different reanalyses suggest that trends in descriptor values are rather a consequence of the reanalysis itself. As answered before, it is difficult to assess which reanalysis is to be trusted, so it is difficult to physically interpret the presence/absence of a trend.

The paragraph that concludes section 4.1 exposes the agreement between reanalyses in terms of variability but the disagreement in terms of trends. We will add a sentence to more explicitly expose the present comments.

*Significance testing: You perform careful and important significance tests for detecting trends/changes in the atmospheric descriptors, which is great, but you don't do the same for the differences in the geopotential anomalies shown in, e.g., Fig. 6. Could you perform such a test and highlight the significant regions? For instance, I doubt that the mentioned reinforcement of the Atlantic anticyclone in winter (L303) in Fig. 6 (top left) is really significant...*

⇒ Thanks for this comment. We will apply a two sample t-test to test the significance of the difference in mean LSC between the two periods, both for Fig. 3 and Fig. 6.

*Investigated domain: I understand the reason for using the study region centered over Central and Southern Europe (rectangle in Fig. 1a), as you are primarily interested in focusing on weather over France. However, I think some of your results for the descriptors might be quite sensitive to the choice of this domain. Did you test this? For instance, I assume increasing the region over whole Europe or even including the North Atlantic might yield results that are more linked to the typical weather regimes in these regions and might thus be easier to interpret. If you do not want to discuss this in too much detail in the manuscript, could you at least extend the anomaly maps in Figs. 1a, 3, and particularly 6 to a larger domain including the main centers of action of the NAO and the further modes of variability (of course by leaving the black rectangles to see the study region)?*

⇒ Thanks for this interesting comment.

In fact, we are more specifically interested in precipitation over the Northern French Alps, located in Southeast France (as exposed in our answer to your first major comment). That was the main reason for considering "regional" circulations over Western Europe rather than considering the whole Europe/North-Atlantic sector. The position and the size of our investigated domain was defined following previous optimization works, notably the one of Raynaud et al. (2017). As discussed in the conclusions of Horton et al. (2012), the domain considered must include the atmospheric flow that drive precipitation (extremes) for the region of interest. In the Northern French Alps, precipitation (extremes) mainly originates from the Atlantic and the Mediterranean (Blanc et al., 2021a; Blanchet et al., 2021; Sodemann and Zubler, 2010). Considering a domain almost centered on the Northern French Alps and including the near Atlantic as well as the Western Mediterranean basin appears a reasonable choice. Moreover, previous studies highlighted the relevance of LSC

for seasonal and extreme precipitation (employing the present atmospheric descriptors) using this domain (Blanc et al., 2021a,b; Blanchet et al., 2018; Blanchet and Creutin, 2020). We do not have investigated small changes in the position of the domain. Geopotential height fields at 500 hPa are quite smooth so we do not think that shifting the domain from 2° will change the results. However, extending the domain to the whole Europe/North-Atlantic sector would most likely lead to different results. At the present time, we haven't downloaded reanalyses data further than 20°W because it was not essential for our study.

Following the major comment of Referee #1, and following your present comment on the considered domain, we have decided to include the study of precipitation in the Northern French Alps in the new version of the paper, in order to discuss the implications of the observed LSC changes for local weather. Considering precipitation in the Northern French Alps will i) support the use of the atmospheric descriptors to derive LSC trends, ii) support the consideration of the domain over Western Europe, and iii) allow for more concrete interpretations of the implications of LSC changes for local weather.

*Method: It took me some time to fully understand the TWS and the derived descriptors (I did not know it before), which I assume other readers might experience, too. Although your schematic in Fig. 1b is very helpful, there are some methodological details you could add to the text. For instance, on L176: How exactly do you compute the TWS between two reanalyses when referring to Eq. 1? What are the days tk and tk' in this context? Do you go through each day (for instance in a specific season) and compute the corresponding TWS between the field of reanalysis 1 and reanalysis 2, which gives you one TWS value for each day, which you then average over the season? This might be a trivial question to you, but it is not obvious from considering Eq. 1, which refers to a day k and another day k'. Furthermore, relating to Fig. 4: Again, I am not sure if I completely understand the details here. Let me explain how I understand it: for the celerity of a specific season between 1850 and 2010 (e.g., for 20CR), you go through each day k during each DJF season and compute the celerity with respect to the previous day k-1. This yields as many celerity values as you have days in this time period. You finally compute a 5-year running mean over this daily celerity time series, which is plotted in Fig. 4 (as anomalies relative to the celerity climatology between 2006 and 2010). Is this correct? Likewise, for the singularity and relative singularity you go through each day and compute the two terms by using a Q of 0.5% of all days available? Are the 0.5% relative to the whole period 1850 to 2010, or do you split it in sub-periods as you mention on L160? And finally, why do you compute the index anomalies against a climatology between 2006 and 2010 only? Would it not be much more robust to use the climatology over the whole investigated period?*

⇒ Thanks for these comments.

As previously answered (first major comment), we will provide a more detailed explanation of the TWS in the method section.

Your interpretation of the construction of Fig. 2 is right. For example: for the comparison of ERA20C with the first member of 20CRv2c, we compute the TWS between the 01/01/1900 of ERA20C and the 01/01/1900 of the first member of 20CRv2c (considering 500 hPa geopotential height fields). We do the same for each day from 01/01/1900 to 31/12/2010. Then we compute the seasonal mean based on the daily score. We will add such an example to ease understanding.

Your interpretation of the construction of Fig. 4 is also right. We will add a sentence in the method section in order to more clearly expose how our descriptors are constructed.

For the singularity and relative singularity, we consider the 0.5% closest days within the period 1950-2010 only, as it is common to the three reanalyses (as exposed l.157-161).

In Fig. 4, we compute anomalies with respect to the 2006-2010 period only to ease comparison. When anomalies are computed with respect to the whole period, the different signal are translated vertically in the plot, which makes the comparison of the reanalyses more difficult. Overall, we mostly interpret Fig. 4 in terms of descriptors trends and variability rather than in their absolute value, so the period of reference has no importance.

**Minor comments**

- *Title: I would write "in Western European large-scale circulation", but I'm not fully sure.*

  ⇒ The expression "Western Europe large-scale circulation" has been used in Blanc et al. (2021a) and Blanc et al. (2021b), so we prefer keeping this expression.

- *L20: You could slightly extend this sentence in the sense that the LSC provides the general dynamical setting for various local weather phenomena, not just via the airflow towards a region but also via modifying stability and moisture availability.*

  ⇒ We will extend this sentence, thanks for this comment.

- *L21: Replace "Over the large scale" with "The"*

  ⇒ Will be done.

- *L23: This sounds a bit odd, because the further (second, third...) modes of variability are also active throughout the year, i.e., not just the NAO.*

  ⇒ Among the four low-frequency atmospheric patterns found in Barnston and Livezey (1987) over Eurasian/North-Atlantic region, the NAO was the only pattern found for every month of the year. However, this is indeed true that other modes of large-scale circulation variability can be obtained in summer applying an empirical orthogonal function to pressure fields (Folland et al., 2009). We will remove this part of the sentence.

- *L25: "by intensifying westerlies"*

  ⇒ Will be done.

- *L25-27: This is an important point, that additional modes of variability are needed to explain surface weather modulation particularly in Central Europe. You could add some further references here, for instance Pasquier et al. (2019), who investigated the link between precipitation extremes in Europe and a higher number of weather regimes than the classic 4 regimes.*

  ⇒ Thanks for the very interesting reference. We will add this reference in the second paragraph of the introduction that focuses on the link between LSC and extreme precipitation.

- *L28: "associated with wet conditions at the northern flank"*

  ⇒ This sentence will be rewritten.

- *L38: Rather something like "low pressure systems developing over the Atlantic and Iberian Peninsula"?*

  ⇒ Will be done, thanks.

- *L39: "associated with strong southwesterly to southerly flows"?*

  ⇒ The term "associated" is already employed in this sentence and we think that "drive" is adapted as the strength and the direction of the flow is driven by the position and strength of the centers of action, including low pressure systems.

- *L43-45: I would distinguish here between stationary cyclones that lead to persistent moisture transport towards the Alps and thus accumulated precipitation and large-scale situations associated with high instability (for instance induced by upper-level cut-off lows; e.g., Portmann et al., 2021) and thus strong forcing for thunderstorms, which are also crucial for (convective) heavy precipitation.*

  ⇒ Thanks for this very relevant comment. In fact, this paragraph will be removed in the updated version of the manuscript in order to better address the specific context related to precipitation in the Western Alps. However, this is an interesting point that we will keep in mind when dealing with different types of LSC-driven extreme precipitation, including summer extreme precipitation in Central Europe.

- *L49: Remove "long-lasting" as it is already stated with "persistence".*

  ⇒ Thanks. This sentence is part of the paragraph that will be removed. It considers the link between LSC and temperature, whereas the new version will focus on LSC and precipitation. The introduction will be more specific to precipitation in the Western Alps.

- *L51-52: You could for instance add Zschenderlein et al. (2019) who investigated the processes behind European heat waves in detail and climatologically.*

  ⇒ This paragraph will be replaced. Thanks for the interesting reference.

- *L55: "may have been linked to"*

  ⇒ This paragraph will also be replaced.

- *L88: "500 hPa geopotential height ranges from"*

  ⇒ Will be done, thanks.

- *L89: Rather "pressure distribution"?*

  ⇒ It is indeed more concise, thanks.

- *L93: How exactly is the ensemble for the 20CRv2c created, i.e. which kinds of "perturbations" are they based on?*

  ⇒ As far as we understand, perturbations are based on observation uncertainty. An Ensemble Kalman Filter is employed to assimilate observations in the 56 members, which allows considering the evolution of observational uncertainties driven by the evolution of the measuring networks (Compo et al., 2011). We will add a sentence about that in the paper.

- *L96: Could you add one or two sentences about the quality of this reanalysis, for instance during the 19th century? I assume continuous pressure / SST / sea ice observations from ships were rather rare at that time?*

  ⇒ It is true. We will add a reference to Rodrigues et al. (2018), who show that the standard deviation in sea level pressure over the North Atlantic between 20CRv2c members is significantly larger in the beginning of the reanalysis because of the lower number of assimilated observations, therefore impacting the mean member homogeneity over time.

- *L103: "referred to as full-input"*

  ⇒ Will be done.

- *L112 and later: I'm not so convinced about the term "atmospheric influences". As they basically refer to the direction of the flow towards Southern France, could you replace it with something like "flow directions", "flow patterns", "weather patterns", or similar? Also, you might switch 3.1 and 3.2, because you mainly start by analyzing the atmospheric descriptors and only then use the atmospheric influences. If done so, you could also split Fig. 1 into two, with a first figure showing the schematic in current Fig. 1b, and a second figure showing the patterns in current Fig. 1a. Furthermore, I would find it very useful to see the links of these four patterns to well-known North Atlantic-European weather regimes / modes of variability. You could partly achieve this by increasing the plotted domain (see one of my major comments). Furthermore, you could write a few sentences about it or refer to other studies that did.*

  ⇒ Thanks for these comments.

  The atmospheric influences represent an aggregation of an existing weather pattern classification featuring 8 weather patterns (Garavaglia et al., 2010). This aggregation is indeed based on the direction of the flow reaching the Northern French Alps but above all on the origin of the air flow (and therefore the origin of humidity), as said in section 3.1. The main sources of humidity in the Western Alpine region are the Atlantic ocean and the Mediterranean Sea (Sodemann and Zubler, 2010), which lead us to aggregate the weather patterns associated with these two sources. Notably, for the Mediterranean influence, we aggregate the weather patterns Central Depression, South Circulation and East Return that all carry humidity from the Mediterranean Sea (WP7, WP4 and WP6 in Fig. 3). However, they represent quite different flow directions. Thus, we prefer keeping the term "influence" when referring to these aggregated weather patterns.

  You're right about the section order. We will switch section 3.1 with section 3.2, as well as Fig. 1a with Fig. 1b.

  The links between Europe/North-Atlantic weather regimes and NAO have been investigated in previous studies for the four atmospheric influences, and little correlations have been found (Blanc et al., 2021b). We will add a sentence about that when introducing the atmospheric influences. Addressing the link between the Western Europe weather patterns and other known Europe/North-Atlantic modes of variability in a quantitative manner would be possible but it seems out of the scope of the present study. Addressing these links in a qualitative manner is difficult because we focus here on "regional" circulation patterns with quite localized pressure anomalies in comparison to the whole Europe/North-Atlantic.

- *Fig. 1a: Could you increase the wind vectors in Fig. 1a, or make them thinner? It is hard to see their directions. Furthermore, are the names Northeast and Anticyclonic really appropriate, as you primarily consider the influence of the pattern on the flow towards Southern France? Would it not rather be something like Northern (for Northeast) and Northeast (for Anticyclonic)?*

  ⇒ The size of the wind vectors will be increased, thanks for this comment. The names of the weather patterns were established in Garavaglia et al. (2010), according to the absolute airflow

[Figure]

Figure 3: Fig. 3 of Garavaglia et al. (2010). Average geopotential height at 1000 hPa of the eight WP (A) and the ratio of the mean WP to global mean precipitation (B). The arrows indicate the atmospheric flow of low layers.

at 1000hPa (arrows in Fig. 3A). Furthermore, the Northeast and Anticyclonic influences will no longer be considered in the present version of the paper as they are weakly related to precipitation in the Northern French Alps (Blanchet et al., 2021; Blanc et al., 2021a,b).

- *L120: "trends being poor" sounds odd – you should rephrase to "small".*

  ⇒ Will be corrected.

- *L178: "to allow for the computation"*

  ⇒ Will be corrected.

- *L162: Why do these three measures only focus on shape? As far as I understand the TWS, it considers the (sum of the) pressure difference of each grid point with respect to each other grid point in a certain domain, and thus indirectly also accounts for differences in pressure magnitudes, right?*

  ⇒ The TWS is actually a normalized score: the differences in zonal and meridional gradients at a given grid point between two days are normalized by the absolute maximum of the two gradients at this grid point. As said earlier: TWS between day $k$ and day $k'$ is a normalized sum of differences in meridional and zonal gradients at all gridpoints over Western Europe between the 500 hPa geopotential height field of day $k$ and day $k'$. Therefore it measures a normalized difference in geopotential shape, i.e. a difference in shape that is independent of the absolute geopotential height nor of the absolute pressure gradient over the region of study. We will add such explanations when introducing the TWS.

- *L166: I am a bit confused about the notation in Eq. 5: Does $max_j$ stand for the maximum value of all grid points $j$ within the rectangle domain? So MPD is the difference of the maximum pressure and minimum pressure (one grid point each) on a certain day $k$?*

  ⇒ You're right: $max_j$ is the maximum geopotential height value of all grid points within the domain. We will remind that $j$ corresponds to a grid point.

- *L168: I assume the weak relation of your MPD to the NAO is simply because your rectangular domain (Fig. 1a) is too small to include the centers of action of the NAO, right? So, if you would increase your domain enough a large MPD would start to coincide with a strong positive or negative NAO, right (see also one of my major comments)?*

  ⇒ You're right, increasing the size of our domain to the whole Europe/North-Atlantic sector would probably make the MPD being more related to the NAO. However, as we will focus in the new version of the paper on precipitation in the Northern French Alps, the MPD is more relevant than NAO, as shown in Blanc et al. (2021b). The central position of the Northern French Alps in Western Europe appears to explain the small correlation with NAO (Bartolini et al., 2009).

- *L170: What exactly do you mean with "per season"? Do you compute all four atmospheric predictors (cel, sing, rsing, mpd) on a daily basis and you then average them over the seasons, or do you compute the atmospheric predictors for the already averaged seasonal geopotential fields? I guess it must be the former, but you should specify this.*

  ⇒ Thanks for this comment. Indeed we compute the four descriptors over each days, and then we arrange the obtained descriptor values per season. We will precise this in the method section.

- *L183: At first sight, it seems counter-intuitive that the geopotential fields agree less well in summer than in winter, considering the generally larger baroclinic activity in the investigated region in winter probably associated with a larger potential for "errors" in the reanalysis. Do you have an idea why they agree less well in summer? Could it be associated with (smaller-scale) convection?*

  ⇒ The TWS focuses on normalized geopotential shapes. In winter, the larger baroclinity activity produces much defined shapes, which facilitate the global representation of the shape in reanalyses. In summer, geopotential shapes are less defined, producing larger differences in shapes among reanalyses. We will discuss this point when interpreting Fig.2.

- *L188: Why are the shapes more similar between 1850 and 1900 than between 1900 and 1950? I would have assumed that the shapes become generally less similar the fewer observations are available (i.e., the further back you go in time), but this is not the case. Is it because you generally increase the degrees of freedom from 1900 on, with the introduction of further types of observations and the 4D-Var? Can you explain?*

  ⇒ This sentence was not clear enough, sorry. We were referring to the differences within 20CR members from 1850 to 1900 that were smaller than differences between 20CR and ERA20C from 1900 to 1950. This looks obvious as differences are larger when comparing different reanalyses products. Taking the 20CR reanalysis alone, the differences in geopotential shapes between members indeed decrease over time. We will rewrite this sentence in order to make it clearer.

- *L194: Did you plot the same as in Fig. 3 but with the absolute fields (i.e., 4 fields for each season: 20CR late-century, 20CR early-century, ERA20C late-century, ERA20C early-century)? If yes, could you show them in your response?*

  ⇒ Fig. 4 to 7 represent such maps. Overall, this is difficult to detect differences in the mean flow both between periods and reanalyses. We can nevertheless note that in general, an increase in 20CR geopotential height is observed between periods with warm colors reaching further north over the whole European sector. This is only the case for ERA20C in Southern Europe, while no changes or decreasing geopotential heights are observed in Northern Europe.

  Fig. 8 allows for a better representation of the differences between the two reanalyses. This corresponds to the updated version of Fig.3 in the paper. Over the period 1900-1930, 20CRv2c features a larger meridional pressure gradient than ERA20C, with larger 500 hPa geopotential height over Southern and Eastern Europe but lower 500 hPa geopotential height over Northwestern Europe, especially in winter. This pattern is reversed over the period 1970-2000, with 20CRv2c featuring larger 500 hPa geopotential height in Northern Europe than ERA20C. Overall, we can note that the shape of the geopotential height differences over Western Europe are more pronounced over the period 1900-1930 than over 1970-2000, and they are also more pronounced in summer than in the other seasons, in accordance with Fig.2 of the paper.

- *L210: Does this mean that 20CR should only be used with caution for studying any climate change / trends over the past 170 years, as it misses a dynamically very relevant change of meridional pressure gradients over the Atlantic-European region (assuming that ERA5 is the best product we have, which is a reasonable assumption)?*

[Figure]

Figure 4: Mean 500 geopotential height for the first member of the 20CR reanalysis (top) and the ERA20C reanalysis (bottom) over the period 1900-1930 (left) and 1970-2000, in winter.

[Figure]

Figure 5: Same as Fig. 4, for spring.

[Figure]

Figure 6: Same as Fig. 4, for summer.

[Figure]

Figure 7: Same as Fig. 4, for autumn.

[Figure]

Figure 8: Updated Fig.3 of the paper. 500 hPa geopotential height difference (in meters) between the composites of i) the first member of 20CRv2c and ERA20C over the period 1900-1930 (first line), ii) the first member of 20CRv2c and ERA20C over the period 1970-2000 (second line), iii) the first member of 20CRv2c over the period 1970-2000 and 1900-1930 (third line), and iv) ERA20C over the period 1970-2000 and 1900-1930 (fourth line), per season. The dots indicate significant differences between the two periods according to a two sample t-test (pvalue<5 %). The Western Europe region over which the atmospheric descriptors are computed is represented by the black rectangle.

⇒ The fact that different LSC changes are found in 20CR and ERA20C cannot lead us to a conclusion of that sort, as comparing reanalyses with an independent data set of observation is not in the scope of the paper. ERA5 is probably the best product as it uses more data, but it is only available after 1950. Thus, we cannot validate the LSC changes observed between the early and the late 20th century in 20CR and ERA20C using ERA5.

- *L217-219: I find it intuitive that the higher resolution and "more details" in ERA5 generally increase the MPD compared to the other reanalyses, but I do not find it obvious for the other descriptors. Wouldn't a higher celerity for instance mean generally less persistent atmospheric states? How do you explain this? Is it because you get more smaller-scale features in ERA5 which tend to be less persistent? Likewise, does the much higher relative singularity in ERA5 imply much more rare/extreme atmospheric states in ERA5 compared to the other reanalyses? Why is this – due to the same reason as above? I think it would be helpful for the reader if you could discuss this a bit.*

  ⇒ Thanks for this comment. As discussed before, the differences in descriptor trends between reanalyses in Fig. 4 can hardly be interpreted in terms of physics, and this is also the case for differences in absolute descriptor values. The observed differences in absolute descriptor values could rely on the different spatial resolution or on the assimilation of observations at different levels in the atmosphere. This makes it difficult to interpret that ERA5 produces more persistent or more rare LSC than the other reanalyses. Furthermore, what can be said about absolute values in Fig. 4 is that i) the higher resolution of ERA5 and ii) the assimilation of upper-air and satellite observation could lead to more detailed geopotential shapes. This could lead to i) larger absolute MPD, and ii) weaker absolute resemblance between shapes because more smaller-scale features are represented. However, we cannot interpret that these smaller-scale features are less persistent in ERA5 than in the other reanalyses, as they are probably not represented in the lower resolution reanalyses.

- *L274: "according to ERA5" or something similar*

  ⇒ Will be done, thanks.

- *Figs. 5, 7, and 8: These are nice figures, but they contain a lot of information. To simplify slightly, would it make sense to replace the classic box plots (right and left of the difference) with violin plots but remove the difference plots instead? You could even make the blue and red violins transparent and overlay them, which would allow directly seeing the absolute values and differences. This would reduce the figure from currently 48 boxes to 16. Could you test this and check whether it is useful?*

  ⇒ Thanks for this comment. This point has been widely discussed with the co-authors when writing the manuscript. Actually, boxplots were our first choice, then boxplots have been replaced by violin plots in order to better show changes in the whole descriptor distribution. Fig. 9 illustrates such a graph for Mediterranean circulations (same as Fig. 8 of the paper). As we can see, violin plots do not allow for a clear comparison of descriptors distribution between the two periods. That's the reason why we introduced a difference in descriptor distribution. Overlying violins appears as a good idea to better see the differences, but then boxplots would be missing in the graph (they cannot be superimposed). We think that boxplots are useful in the graph as this is one of the most common representation of distribution in research article. Therefore, we propose to keep the current version of the

[Figure]

Figure 9: Same as Fig. 8 of the paper, but with violin plots.

graphs. Note however that they only represent two figures in the present version of the paper as the Anticyclonic influence will no longer be considered (Fig.5 will be removed).

- *Fig. 5: I'm not sure if I understand correctly: You say that the percentiles are defined with respect to all days within 1950 and 2019. So, shouldn't the sum of the blue and red bars consequently give a distribution that is somewhat centered around the 50th percentile on the y-axis? For many descriptors, such as the celerity in winter or summer, however, this does not seem to be the case. Why is that?*

  ⇒ Percentiles are computed with respect to all days within 1950 and 2019 belonging to the same atmospheric influence, as detailed in the caption of Fig. 5. The boxplots would have been centered around the 50th percentile if considering the whole year, but some descriptors feature a seasonal signal (Fig. 10, Blanc et al., 2021a), so some medians differ from 50%.

- *L296: "meaning new anticyclonic patterns are less explored over the present period" – this sounds a bit strange, although I know what you mean referring to the singularity. It sounds like we had completely different atmospheric patterns in the earlier period, which do not exist anymore. Can you rephrase somehow?*

  ⇒ As previously said, the study of the Anticyclonic influence will be removed in the new version of the paper as it is not relevant for precipitation in the Northern French Alps.

- *L300: "associated with high pressure"*

  ⇒ Same as the previous comment.

- *L310: "reproducibility of Atlantic circulations" sounds a bit like a model that fails to reproduce these patterns, although we are dealing with reanalysis. Could you rephrase somehow?*

  ⇒ Will be done, thanks.

- *Figs. 9 and 10: These two figures appear out of a sudden in Section 4.2.3, without introducing the reasons behind showing them (for instance, why only for summer and autumn, and the Mediterranean circulation?). Furthermore, the description in the caption of Fig. 10 is hard*

[Figure]

Figure 10: Fig. 7 of Blanc et al. (2021a). Interannual median (blue), 0.1 and 0.9 quantiles (grey) of the celerity, singularity, relative singularity and MPD for the period 1950–2017.

*to understand and I had to read it several times. In general, I wonder if these two figures are really needed or whether you could simply state some of the most important changes in, for instance, the occurrence frequency of the most extremely stationary states? Although the idea behind Fig. 9 is good, for instance, I find it hard to grasp a message from looking at this figure, as the two distributions are overall rather similar. Removing these two figures would further help to reduce the content of the rather long manuscript...*

⇒ Thanks for this comment. In the new version of the paper, Fig. 9 and 10 will be replaced by figures providing more relevant information on LSC driving precipitation in the Northern French Alps. However, we will keep one sentence related to the percentages previously exposed in Fig. 10.

- *L337-339: What might be the physical reason to obtain such a "seasonal shift" as you call it (considering the fact that the solar cycle, i.e. the main driver of the seasonal cycle, should not be affected by any change)? Isn't it more the northward shift of the storm track that we might see here, which influences the Mediterranean differently during the different seasons because its overall year-round meridional oscillation also changes with the northward shift?*

⇒ Thanks for this comment. You're right, explanations of such a probable seasonal shift are not obvious and we do not have a clear vision on the potential causes. Maybe the northward shift of the storm-track in winter do not allow for pronounced Mediterranean circulations in winter, but this is just one of the possible hypothesis. We will therefore remove the "seasonal shift" suggestion.

- *Conclusions: Coming back to one of my major comments, I miss some more specific implications / ideas for further research in this last paragraph. When reading the conclusions, it is not clear whether your main take-home message should be the usefulness of your method*

*to better understand surface weather extremes in France, or whether it is more about better understanding historic large-scale changes over Europe in general with this special approach and highlighting differences in reanalysis products. To me, it is kind of both, but you should state/distinguish this more clearly. Also, you could again refer some of your results to some other specific studies or ongoing debates. For instance, you could relate your findings about celerity (i.e., stationarity) to the widely discussed (but partly disputed) increase in jet stream waviness due to Arctic amplification, potentially leading to more persistent and thus extreme weather (e.g., Kornhuber & Tamarin-Brodsky, 2021).*

⇒ Thanks for this comment. Regarding the conclusions of section 4.1, we will more clearly highlight that trends in Western Europe LSC before 1950 must be taken with caution when using long-term reanalyses. Regarding the implications of the results of section 4.2, we will more clearly expose the influence of LSC changes on precipitation variability and extremes in the Northern French Alps, as this study will be explicitly carried out in the paper. We will highlight the relevance of the atmospheric descriptors to reach the conclusions of both section 4.1 and 4.2.

Thanks for the reference; the implication of changes in meridional temperature gradient for changes in jet-stream strength and meandering is indeed a fascinating debate. However, changes observed in celerity after 1950 (when the different reanalysis agree) are not the most pronounced. We will focus the discussion on the relevance of other (thermodynamic) variables that allow a more complete explanation of the observed trends in seasonal and extreme precipitation in the Northern French Alps.

[revised manuscript text omitted]

---

## Referee Report (RR1)

**2nd review of "Past Evolution of Western Europe Large-scale Circulation and Link to Precipitation Trend in the Northern French Alps" by Blanc et al. submitted to Weather and Climate Dynamics (WCD-2021-69)**

**General comments**

Thank you for addressing all my comments so thoroughly and providing additional explanations where necessary! I particularly think that the modified/new last part of the results now provides a nice application and synthesis of the trends in atmospheric predictors and circulations to better understand the observed changes in precipitation. I only have a few minor suggestions, which should be considered before the publication of this manuscript.

**Minor comments (line numbers refer to revised manuscript)**

L12-14: "…we show that winter Atlantic circulations (zonal flows) tend to be shifted northward and they become more similar to known Atlantic circulations. Mediterranean circulations tend to become more stationary, more similar to known Mediterranean circulations and associated with stronger centers of action in autumn…" – I know by now that you refer to previous studies when talking about the "Atlantic circulations" and the "Mediterranean circulations". But these two terms sound a bit strange when reading them for the first time in the abstract without any further explanation. Could you briefly describe the patterns when mentioning them in the abstract? "Atlantic circulation" sounds very unspecific, as there is always circulation there. Maybe you can write something like "the Atlantic circulation pattern dominated by a trough over the UK" (and likewise a description for the Mediterranean circulation pattern).

L16: Change to "…we show that these changes…"

L49: Rather "…have pointed out the link…"? (this also appears at other locations in the manuscript)

L66/379: "…responsibility of LSC changes…" sounds a bit strange to me. Maybe write something like "…role of LSC changes…"?

L105-106: Of course, I know what you mean with "…mainly affected by precipitation coming from the Atlantic…", but I would still change the wording here a bit. It is rather the moisture brought in from the Atlantic, while the precipitation is probably formed later, also in interaction with the orography.

L123: Maybe write "…where Adj ranges the set of adjacent 4 grid points…" that it becomes clear you only talk about the two grid points in meridional and zonal direction, respectively (at least this is how I understand it from your manuscript).

L245-247: I apologize but I think I still don't fully understand one aspect in Fig. 5. You write that the lines show the 5-year-running-mean anomalies with respect to the 2006-2010 average. Does this mean that, for instance, the ERA20C line (red) shows the anomaly with respect to the 2006-2010 value of ERA20C? If so, why do all the lines at the end of the time series (2010) have the same value, while the dots between the different reanalysis products show large differences (which you discuss in lines 245-247)? Maybe you need to add a few words to clarify this in the caption.

L352-354: "In summer and autumn, an opposite pattern is observed with a decreasing 500 hPa geopotential height over Northwestern Europe reaching further South in autumn, pointing to a reinforcing of Mediterranean circulations." – I don't understand how the Z500 changes in summer in Fig. 7 (bottom)

can reinforce the Mediterranean circulation pattern shown in Fig. 2b. There is an increase in pressure over Central Europe and thus in between the centers of action of the Mediterranean circulation pattern – how can this intensify the cyclonic circulation centered west of the Iberian Peninsula?

L389: I guess you want to refer to Fig. 6 here but not Fig. 7, right?

L432: I guess you mean "Fig. 13 right"?

L456: Change to "…became more frequent…"

L463: I would rather write: "…the Mediterranean circulations becoming less pronounced and less closely reproduced in winter over the last 30 years…". In the current form, it sounds more like the frequency of the Mediterranean circulation pattern changes, which, however, is not the case as (I think) you mention earlier in the manuscript.

L467-471: It is very important and good that you write the paragraph about the limitations regarding the robustness of the trends. However, it sounds a bit unfortunate, as it kind of questions a large part of your study when written like this. Although you are right that we should not trust the trends too much, I still think your study shows how the atmospheric descriptors in combination with the patterns can (potentially) help to understand changes in precipitation. Maybe this will become even more important for studying future precipitation changes (as you briefly mention, but it could still be highlighted more explicitly that you could apply this method to precipitation changes in GCM projections). So I think you should briefly state this after this "limitation paragraph".

L472-473: Change to "…this evolution is partly relevant..."

Fig. 13: I'm not so sure how helpful this figure is in its current form, as it's hard to disentangle the individual lines. Also, how exactly do you calculate the descriptor values at these individual days? For the celerity, you need two days, right? Do you just use the previous day for each date? Furthermore, when reading your text it seems to me that Fig. 13 is not crucial to understand the trends in extreme precipitation. So I guess you could think of omitting Fig. 13, but I guess it is a matter of taste and I'll leave it up to you to decide.

---

## Author Response (AR2)

We thank Referee #2 for the new feedbacks. Please find below a detailed point-to-point reply to the Referee's comments.

**1 General comments**

*Thank you for addressing all my comments so thoroughly and providing additional explanations where necessary! I particularly think that the modified/new last part of the results now provides a nice application and synthesis of the trends in atmospheric predictors and circulations to better understand the observed changes in precipitation. I only have a few minor suggestions, which should be considered before the publication of this manuscript.*

**2 Minor comments (line numbers refer to revised manuscript)**

- *L12-14: ". . . we show that winter Atlantic circulations (zonal flows) tend to be shifted northward and they become more similar to known Atlantic circulations. Mediterranean circulations tend to become more stationary, more similar to known Mediterranean circulations and associated with stronger centers of action in autumn. . . " – I know by now that you refer to previous studies when talking about the "Atlantic circulations" and the "Mediterranean circulations". But these two terms sound a bit strange when reading them for the first time in the abstract without any further explanation. Could you briefly describe the patterns when mentioning them in the abstract? "Atlantic circulation" sounds very unspecific, as there is always circulation there. Maybe you can write something like "the Atlantic circulation pattern dominated by a trough over the UK" (and likewise a description for the Mediterranean circulation pattern).*

  ⇒ Thanks, we have added a sentence in the abstract to better introduce these terms.

- *L16: Change to ". . . we show that these changes. . . "*

  ⇒ Done, thanks.

- *L49: Rather ". . . have pointed out the link. . . "? (this also appears at other locations in the manuscript)*

  ⇒ Done, thanks.

- *L66/379: ". . . responsibility of LSC changes. . . " sounds a bit strange to me. Maybe write something like ". . . role of LSC changes. . . "?*

  ⇒ Done, thanks.

- *L105-106: Of course, I know what you mean with ". . . mainly affected by precipitation coming from the Atlantic. . . ", but I would still change the wording here a bit. It is rather the moisture brought in from the Atlantic, while the precipitation is probably formed later, also in interaction with the orography.*

  ⇒ You're right, moiture is more relevant than precipitation here, as orography plays a leading role in triggering precipitation.

- *L123: Maybe write "... where Adj ranges the set of adjacent 4 grid points..." that it becomes clear you only talk about the two grid points in meridional and zonal direction, respectively (at least this is how I understand it from your manuscript).*

  ⇒ *Adj* actually refers to every pair of adjacent grid points in the eastern and northern direction in the domain. Let's say that $i$, $j$, and $k$ are three consecutive grid points at the same latitude. We consider two pairs: (i,j) and (j,k). We do the same in the longitudinal direction. In the paper, we have clarified that we refer to gradients in the eastern and northern direction.

- *L245-247: I apologize but I think I still don't fully understand one aspect in Fig. 5. You write that the lines show the 5-year-running-mean anomalies with respect to the 2006-2010 average. Does this mean that, for instance, the ERA20C line (red) shows the anomaly with respect to the 2006-2010 value of ERA20C? If so, why do all the lines at the end of the time series (2010) have the same value, while the dots between the different reanalysis products show large differences (which you discuss in lines 245-247)? Maybe you need to add a few words to clarify this in the caption.*

  ⇒ The anomaly with respect to the 2006-2010 is indeed represented in Fig.5 for each reanalysis. By definition, subtracting the 2006-2010 average value to a given reanalysis makes it centered around 0 in 2006-2010. As we do the same for every reanalysis according to their respective 2006-2010 value, every signal is centered around 0 in 2006-2010. The dots in 2010 simply show the differences in 2006-2010 values between reanalyses. For instance, if the MPD average of ERA5 is 300 m in 2006-2010 and the MPD of ERA20C is 280m in 2006-2010, the dot of ERA20C will be at $-20$ m and that of ERA5 will be at 0 as we arbitrary chose ERA5 as the reference. So the dots represent the translation applied to the signal to bring them together. We have clarified that Fig.5 represents time series anomalies according to their respective 2006-2010 average.

- *L352-354: "In summer and autumn, an opposite pattern is observed with a decreasing 500 hPa geopotential height over Northwestern Europe reaching further South in autumn, pointing to a reinforcing of Mediterranean circulations." – I don't understand how the Z500 changes in summer in Fig. 7 (bottom) can reinforce the Mediterranean circulation pattern shown in Fig. 2b. There is an increase in pressure over Central Europe and thus in between the centers of action of the Mediterranean circulation pattern – how can this intensify the cyclonic circulation centered west of the Iberian Peninsula?*

  ⇒ The larger increase in pressure during summer Mediterranean circulation indeed occurs in between the centers of action. However, the increase in pressure is larger over Central Europe– a region where high pressure anomalies are observed during Mediterranean circulations–than over the near Atlantic–a region where low pressure anomalies are observed during Mediterranean circulations. This reflects an increase in pressure gradient and a reinforcement of Mediterranean circulations. We have rewritten part of this paragraph to better highlight the observed differences in LSC changes between summer and autumn.

- *L389: I guess you want to refer to Fig. 6 here but not Fig. 7, right?*

  ⇒ You're right, thank you.

- *L432: I guess you mean "Fig. 13 right"?*

⇒ You're right, thank you.

- *L456: Change to "...became more frequent..."*

  ⇒ Done, thanks.

- *L463: I would rather write: "...the Mediterranean circulations becoming less pronounced and less closely reproduced in winter over the last 30 years...". In the current form, it sounds more like the frequency of the Mediterranean circulation pattern changes, which, however, is not the case as (I think) you mention earlier in the manuscript.*

  ⇒ Done. Indeed it is a better formulation. Nevertheless, changes in Mediterranean circulations characteristics reflect in a way changes in the frequency of specific pattern within this atmospheric influence.

- *L467-471: It is very important and good that you write the paragraph about the limitations regarding the robustness of the trends. However, it sounds a bit unfortunate, as it kind of questions a large part of your study when written like this. Although you are right that we should not trust the trends too much, I still think your study shows how the atmospheric descriptors in combination with the patterns can (potentially) help to understand changes in precipitation. Maybe this will become even more important for studying future precipitation changes (as you briefly mention, but it could still be highlighted more explicitly that you could apply this method to precipitation changes in GCM projections). So I think you should briefly state this after this "limitation paragraph".*

  ⇒ Thanks for this comment. We have reorganized the last paragraph of the conclusions and we have more clearly exposed that our methodology could be applied to GCM outputs.

- *L472-473: Change to "...this evolution is partly relevant..."*

  ⇒ Done, thanks.

- *Fig. 13: I'm not so sure how helpful this figure is in its current form, as it's hard to disentangle the individual lines. Also, how exactly do you calculate the descriptor values at these individual days? For the celerity, you need two days, right? Do you just use the previous day for each date? Furthermore, when reading your text it seems to me that Fig. 13 is not crucial to understand the trends in extreme precipitation. So I guess you could think of omitting Fig. 13, but I guess it is a matter of taste and I'll leave it up to you to decide.*

  ⇒ It is indeed hard to disentangle the indivudual lines, but all we interpret from Fig.13 is the linear regressions applied to the different signals. The descriptor values shown here are based on the daily descriptor values computed from the beginning of the paper. For the celerity, Fig.13 indeed represents the TWS between the day of annual maximum of precipitation and the day before. We agree that Fig.13 is probably not the most important figure of the paper, but still it fuels the discussion on the role of LSC changes for extreme precipitation trends. We have then decided to keep this figure.